# DNA methylation at retrotransposons protects the germline by preventing NRF1-mediated activation

Jessica Leismann [1,6 ✉], Styliani-Eirini Kanta [1,6], Ishita Amar[1], Anna Szczepińska [1], Monika Mielnicka [2], Giuseppe Petrosino [1], Anke Busch[1], Marion Scheibe[3], Pengxiang Wang [4], Yuan Wang [5], Falk Butter[3], Matthieu Boulard [2] & Joan Barau [1 ✉]

## Abstract

**Silencing evolutionary young retrotransposons by cytosine DNA methylation is essential for spermatogenesis, as failure to methylate their promoters leads to reactivation, meiotic failure, and infertility. How retrotransposons reactivate in the absence of DNA methylation is poorly understood. We show that upon defective DNA methylation, distinct retrotransposon families display unique expression patterns and chromatin landscapes during mouse spermatogenesis. We find that their reactivation in meiotic spermatocytes correlates with the loss of bivalent H3K4me3-H3K27me3 chromatin marks. Through proteomics and chromatin profiling, we identify NRF1 as a DNA methylation-sensitive transcription factor that transactivates unmethylated retrotransposons. Conditional germline knockout of *Nrf1* in the absence of DNA methylation rescues the silencing of the most mutagenic retrotransposon in mice, namely Intracisternal A-particle or IAP. Our findings reveal that chromatin modifications together with a DNA methylation-sensitive transcription factor regulate retrotransposon expression in the absence of DNA methylation in spermatogenesis, revealing a mechanism by which retrotransposons proliferate in the germline after evading DNA methylation-based silencing.**

**Keywords** Transposable Elements; Germline; DNA Methylation; NRF1; Chromatin Modifications
**Subject Categories** Chromatin, Transcription & Genomics; Development

## Introduction

Eukaryotic genomes are littered by copies of genetic elements of parasitic origin known as transposable elements (TEs). TEs can be beneficial for the host organism, but TE activity also poses an immediate risk to genomic stability (Chuong et al, 2017). To mitigate this permanent threat, mammals have evolved various mechanisms to repress TEs, and one of the most prominent is cytosine DNA methylation that imposes strong silencing and is propagated across cell divisions (Yoder et al, 1997). In the germline, efficient TE silencing is particularly critical, as new TE insertions can be inherited by the next generation. Paradoxically, DNA methylation undergoes a nearly complete reprogramming event in early developing germ cells of mammals, exposing the germline genome to the threat of TE activity (Seisenberger et al, 2012; Monk et al, 1987). In mouse fetal germ cells, this threat is counteracted by the piRNA pathway, which recognizes active TEs and guides de novo DNA methylation to their promoters through the specialized methyltransferase DNMT3C and its catalytically inactive cofactor DNMT3L (Bourc'his and Bestor, 2004; Barau et al, 2016; Jain et al, 2017). Disruption of this pathway and the resulting loss of DNA methylation releases young TEs from silencing, leading to meiotic failure and male sterility (Bourc'his and Bestor, 2004; Barau et al, 2016; Aravin et al, 2008). Specifically, the long interspersed nuclear elements-1 (LINE1, L1) are highly upregulated at meiosis in the absence of DNA methylation (Zamudio et al, 2015; Barau et al, 2016). In contrast, the endogenous retrovirus (ERV) type known as Intracisternal A-particle (IAP) is reported to be activated as early as the undifferentiated spermatogonia stage (Vasiliauskaite et al, 2018). This varied time window of TE reactivation upon loss of DNA methylation indicates that mechanisms beyond DNA methylation influence the expression of different TE families. One such regulatory mechanism involves repressive histone modifications such as H3K9me2, H3K9me3, and H3K27me3, which play a crucial role in controlling TE activity when DNA methylation is absent (Di Giacomo et al, 2014; Zamudio et al, 2015; Walter et al, 2016).

Furthermore, TEs can be silenced by repressive complexes binding methylated DNA (Rowe et al, 2010; Boulard et al, 2020) or by inhibition of the binding of DNA methylation-sensitive transcription factors (TFs) (Kaluscha et al, 2022). TFs bind to specific DNA sequences and influence gene expression patterns (Maston et al, 2006). Their access to the DNA can be further regulated by epigenetic mechanisms, including nucleosome positioning, histone modifications, and DNA methylation (Inukai et al, 2017). While DNA methylation at cytosines within CpG dinucleotides can inhibit binding of some TFs, others, such as REST, have been shown to bind methylated regions and promote their

[1]Institute of Molecular Biology, Mainz, Germany. [2]European Molecular Biology Laboratory, EMBL, Rome, Italy. [3]Friedrich-Loeffler-Institut, Greifswald, Germany. [4]Department of Physiology, Navy Medical University, 200433 Shanghai, China. [5]Department of Animal Science, Michigan State University, East Lansing, MI, USA. [6]These authors contributed equally: Jessica Leismann, Styliani-Eirini Kanta. ✉E-mail: jessica.leismann95@gmail.com; joan.barau@gmail.com

demethylation (Tate and Bird, 1993; Feldmann et al, 2013; Stadler et al, 2011; Yin et al, 2017; Rimoldi et al, 2024). Moreover, TF activity can be modulated by posttranslational modifications as well as by the presence of cooperative binding partners (Everett et al, 2010; Long et al, 2016). Several TFs have been implicated in TE regulation across various cell types, but most TFs exhibit cell-type specificity, and no universal activator of TEs is known (Karttunen et al, 2023; Hermant and Torres-Padilla, 2021; Sundaram et al, 2014). For example, CREB1 has been shown to bind unmethylated IAPs in differentiated neurons (Kaluscha et al, 2022; Sakashita et al, 2023). However, only a few examples of TFs that bind directly to TE promoters in germ cells have been identified (Otsuka et al, 2023), and their contribution to the activation of TEs in germ cells is unknown.

Previous studies have focused on the reactivation of TEs in germ cells following widespread loss of DNA methylation (Bourc'his and Bestor 2004; Pastor et al, 2014; Zamudio et al, 2015; Vasiliauskaite et al, 2018). In this work, we used the Dnmt3C knockout ($Dnmt3C^{KO/KO}$) mouse model (Barau et al, 2016; Jain et al, 2017) as a paradigm for defective germline DNA methylation restricted to TE promoters. DNMT3C is guided by the piRNA pathway and specifically methylates the promoters of retrotransposons that are potentially active in germ cells (Barau et al, 2016; Zoch et al, 2024). Key to our work, $Dnmt3C$ mutant germ cells offer a near physiological landscape of global genomic DNA methylation, with defects restricted to the regulatory sequences of TE promoters (Barau et al, 2016). We used the $Dnmt3C$ mutants to closely follow TE activity by a combination of immunostaining, western blotting, and RNA sequencing, uncovering previously hidden patterns of expression that appeared to follow the stages of spermatogenic development. We also profiled several active and repressive histone modifications using CUT&Tag in isolated germ cells, uncovering novel shifts in potential regulatory chromatin that correlated with TE expression patterns in spermatogenesis. Finally, we identify NRF1 as a DNA methylation-sensitive TF required for the transcription of IAPs.

## Results

### Young retrotransposons become expressed prior to meiosis in $Dnmt3C^{KO/KO}$ spermatogenesis

To gain deeper insight into the impact of local DNA hypomethylation on the activity of young retrotransposons throughout spermatogenesis, we assessed their expression in $Dnmt3C^{KO/KO}$ germ cells compared to wild-type. The first wave of spermatogenesis in mice is synchronized, allowing us to track distinct developmental stages of spermatogenesis by collecting testes at precise embryonic and postnatal time points (Fig. EV1A). Specifically, we collected testes at embryonic day (E)15.5, where germ cells are in the prospermatogonia stage before MIWI2 becomes enriched in the nucleus and therefore preceding piRNA-directed DNA methylation (Aravin et al, 2008; Zoch et al, 2020). To reference a prospermatogonia stage when piRNA-directed DNA methylation is ongoing, we collected testes at E18.5 (Barau et al, 2016). At postnatal day (P)5, complete DNA methylation patterns are established in wild-type and germ cells are in the spermatogonial stem cell and spermatogonia stages. By P10, the first wave of

meiosis is evident, with most germ cells in meiosis by P15. P20 was chosen to capture a timepoint in development, when meiotic cells are past the developmental stage of arrest in $Dnmt3C^{KO/KO}$ mutants, while at P30 the germ cells have completed the first wave of meiosis (Ernst et al, 2019). Altogether, these stages encompass male germ cells throughout the genome-wide reprogramming of DNA methylation—spanning the period before and during DNMT3C activity, as well as the later stages when DNA methylation is maintained, both in wild-type and $Dnmt3C^{KO/KO}$ conditions (Fig. EV1B).

As a proxy for young retrotransposon activity, we first examined LINE1 (L1) ORF1 protein (L1-ORF1p) expression across these stages using immunofluorescence on testes cryosections (Figs. 1A and EV1C,D). As expected, L1-ORF1 was detected in prospermatogonia at E15.5 in both wild-type and $Dnmt3C^{KO/KO}$, before de novo methylation (Molaro et al, 2014; Barau et al, 2016). From E18.5 onwards, we observed L1 silencing in wild-type cells as expected. In contrast, L1-ORF1p expression persisted until P30 in $Dnmt3C^{KO/KO}$ cells, reflecting the failure to transcriptionally silence retrotransposons due to the lack of DNMT3C, resulting in DNA methylation defects (Barau et al, 2016). Notably, we also observed L1-ORF1 protein in $Dnmt3C^{KO/KO}$ spermatogonia at P5, prior to meiosis (Fig. 1A). This is in contrast with previous observations in several mutants of the piRNA pathway, such as $Mael^{KO/KO}$ (Soper et al, 2008), $Dnmt3L^{KO/KO}$ (Zamudio et al, 2015), and $Mili^{KO/KO}$ (Di Giacomo et al, 2013), where L1-ORF1 protein was not detected before meiosis. The differences observed in our study could be attributed to the $Dnmt3C^{KO/KO}$ mouse model, which only affects de novo methylation at young TE promoters. As previously described in $Dnmt3L^{KO/KO}$, $Mili^{KO/KO}$, and $Dnmt3C^{KO/KO}$ mutants, we observed a drastic increase of L1 activity once germ cells entered meiosis, starting at P10 (Figs. 1A and EV1C,D) (Zamudio et al, 2015; Di Giacomo et al, 2013; Soper et al, 2008).

Western blot analysis confirmed these findings, revealing the presence of L1-ORF1 in all $Dnmt3C^{KO/KO}$ testes extracts (Fig. 1B). A clear difference in L1-ORF1 signal between $Dnmt3C^{KO/KO}$ and wild-type was first evident by E18.5. In $Dnmt3C^{KO/KO}$ at stage P5, the signal of L1-ORF1 appears slightly reduced compared to E18.5, in agreement with the quantification of L1-ORF1 signal assessed by immunofluorescence (Fig. EV1D). At P15, L1-ORF1 intensity increased in comparison to other stages, with the strong signal observed in $Dnmt3C^{KO/KO}$ reflecting the known upregulation of transposons in meiosis (Zamudio et al, 2015). In addition, we examined the expression of IAPs by western blot using an antibody against IAP-POL across the same developmental stages and observed a similar pattern of expression as for L1s. However, a clear difference in IAP-POL band intensity between wild-type and $Dnmt3C^{KO/KO}$ became only apparent by P5 and increased by P15 (Fig. 1B).

Next, we performed RNA sequencing on FACS-isolated postnatal premeiotic spermatogonia and meiotic spermatocytes from wild-type and $Dnmt3C^{KO/KO}$ mice to characterize the transcriptional activity of retrotransposon families in response to the local loss of promoter DNA methylation (Fig. 1C–E; Appendix Fig. S1A–D). In $Dnmt3C^{KO/KO}$ spermatogonia, we detected a marked transcriptional upregulation of the most active ERVK subclass (Dewannieux et al, 2004) and their associated LTRs, namely IAP (LTR1, LTR2) and MMERVK10C (LTR10C). Upregulation was also observed for L1 families L1MdTf, L1MdA, L1MdGf (Fig. 1C).

## A

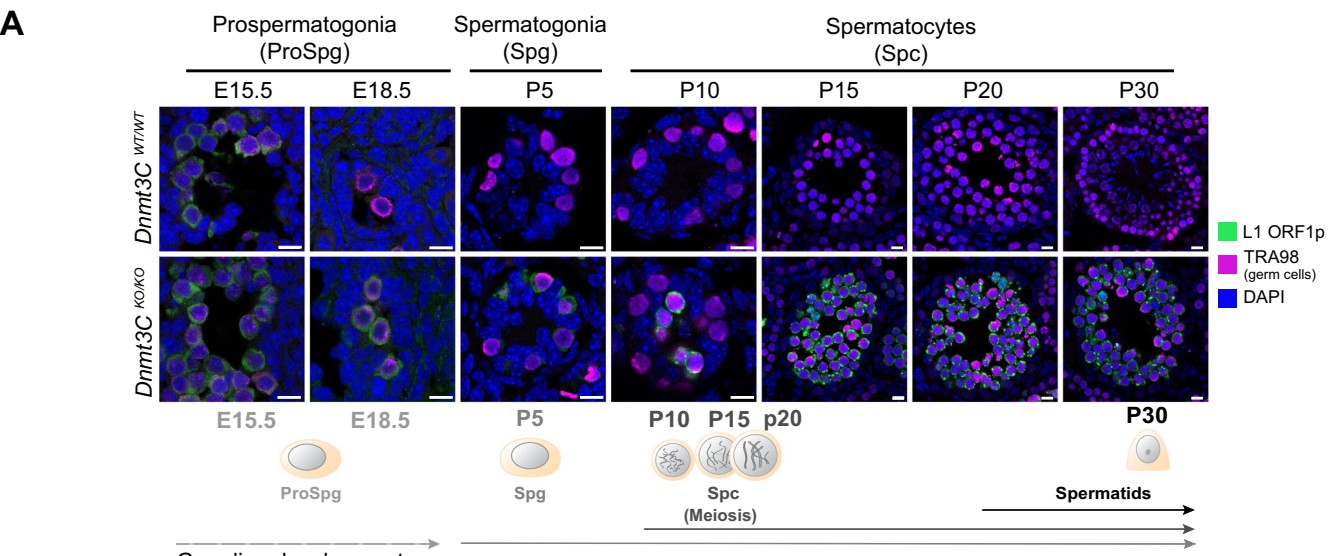

Prospermatogonia (ProSpg) Spermatogonia (Spg) Spermatocytes (Spc)

E15.5 E18.5 P5 P10 P15 P20 P30

$Dnmt3C^{WT/WT}$

$Dnmt3C^{KO/KO}$

L1 ORF1p
TRA98 (germ cells)
DAPI

E15.5 E18.5 P5 P10 P15 p20 P30

ProSpg Spg Spc (Meiosis) Spermatids

Germline development

## B

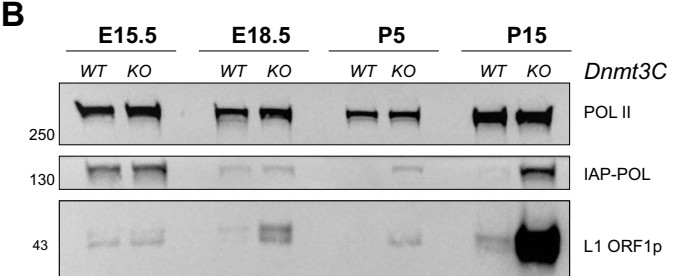

E15.5 E18.5 P5 P15

WT KO WT KO WT KO WT KO $Dnmt3C$

250 — POL II

130 — IAP-POL

43 — L1 ORF1p

## D

### Spermatocytes $Dnmt3C^{KO/KO}$ vs $Dnmt3C^{WT/WT}$

downregulated in $Dnmt3C^{KO/KO}$ upregulated in $Dnmt3C^{KO/KO}$

● NS ● P−adjust <= 0.01

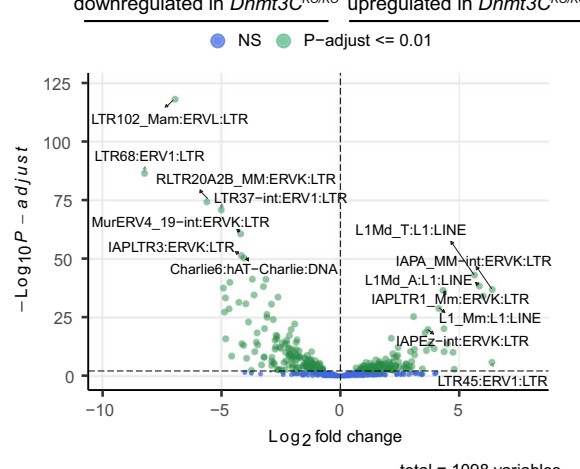

LTR102_Mam:ERVL:LTR
LTR68:ERV1:LTR
RLTR20A2B_MM:ERVK:LTR
LTR37-int:ERV1:LTR
MurERV4_19-int:ERVK:LTR
IAPLTR3:ERVK:LTR,
Charlie6:hAT-Charlie:DNA
L1Md_T:L1:LINE
IAPA_MM-int:ERVK:LTR
L1Md_A:L1:LINE
IAPLTR1_Mm:ERVK:LTR
L1_Mm:L1:LINE
IAPEz-int:ERVK:LTR
LTR45:ERV1:LTR

$-\mathrm{Log}_{10}P-adjust$

$\mathrm{Log}_2$ fold change

total = 1098 variables

## C

### Spermatogonia $Dnmt3C^{KO/KO}$ vs $Dnmt3C^{WT/WT}$

downregulated in $Dnmt3C^{KO/KO}$ upregulated in $Dnmt3C^{KO/KO}$

● NS ● P−adjust <= 0.01

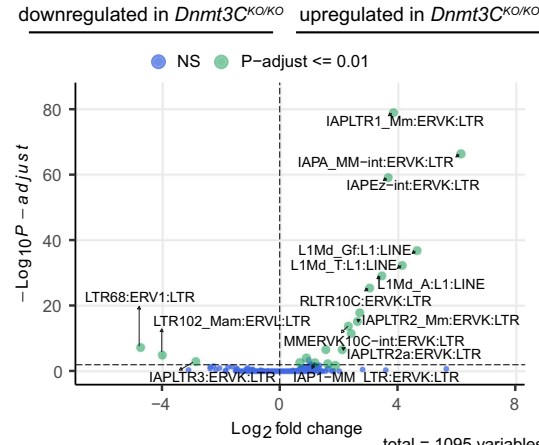

IAPLTR1_Mm:ERVK:LTR
IAPA_MM-int:ERVK:LTR
IAPEZ-int:ERVK:LTR
L1Md_Gf:L1:LINE
L1Md_T:L1:LINE
L1Md_A:L1:LINE
LTR68:ERV1:LTR
RLTR10C:ERVK:LTR
IAPLTR2_Mm:ERVK:LTR
LTR102_Mam:ERVL:LTR
MMERVK10C-int:ERVK:LTR
IAPLTR2a:ERVK:LTR
IAPLTR3:ERVK:LTR IAP1-MM_LTR:ERVK:LTR

$-\mathrm{Log}_{10}P-adjust$

$\mathrm{Log}_2$ fold change

total = 1095 variables

## E

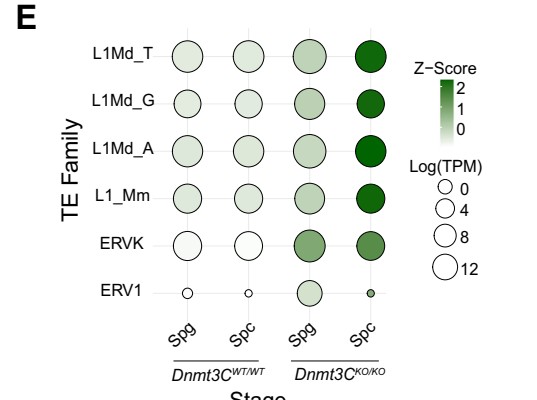

TE Family: L1Md_T, L1Md_G, L1Md_A, L1_Mm, ERVK, ERV1

Spg Spc Spg Spc
$Dnmt3C^{WT/WT}$ $Dnmt3C^{KO/KO}$

Stage

Z-Score: 2, 1, 0

Log(TPM): 0, 4, 8, 12

**Figure 1. Young retrotransposons are expressed prior to meiosis in *Dnmt3C^{KO/KO}* spermatogenesis.**

(A) Representative immunostaining from biological duplicates of LINE1-ORF1p and TRA98 (germ cell marker) on cryosections of wild-type (WT) and *Dnmt3C^{KO/KO}* testes at specific developmental time points. For visualization purposes, the brightness and contrast were adjusted for DAPI and TRA98, but not for L1-ORF1p. Illustration of germ cell development progression at the bottom. E15.5 and E18.5 represent prospermatogonia cell stages (ProSpg). P5 corresponds to a postnatal stage with germ cells at spermatogonia stages (Spg). Spermatocytes (Spc) begin at P10, and spermatids start to appear at P20. Scale bars, 10 μm. (B) Representative Western blot analysis of LINE1-ORF1p, IAP-POL and RNA polymerase 2 (POL II) as a loading control from wild-type and *Dnmt3C^{KO/KO}* testes at the indicated developmental time points. The blot was developed using Pico chemistry. The experiment was performed in biological duplicates. (C) Volcano plots showing DESeq2 results with -log$_{10}$P-adjusted values from the Wald test and log2 fold changes from three biological replicates of differentially expressed TE-transcripts between wild-type and *Dnmt3C^{KO/KO}* sorted germ cells at spermatogonia stages. The display of TE names follows the pattern TEname:Family:Class (D) as (C) at spermatocyte stages. (E) Bubble chart displaying average Z-score and log(TPM) of expression values from three biological replicates of wild-type and *Dnmt3C^{KO/KO}* (Spg and Spc) for TE families that were differentially expressed in Spg and Spc. Source data are available online for this figure.

Our RNA-sequencing data reveal that L1s are upregulated in spermatogonia, providing orthogonal validation of L1-ORF1p detection by stage-specific western blot and immunofluorescence. While IAP expression at this stage has been previously reported in the context of a defective piRNA pathway (Vasiliauskaite et al, 2018), our findings demonstrate that L1s are also active at this stage. Of note, in wild-type spermatocytes, we observed upregulation in several TE families, particularly populated by ancient TEs (Dataset EV1). This upregulation in wild-type spermatocytes likely reflects differences in germ cell composition of wild-type samples compared to *Dnmt3C^{KO/KO}*, where an arrest occurs at the pachytene stage of meiosis (Barau et al, 2016). In wild-type spermatocytes, the sorted population also includes a later developmental stage representing a different transcriptional state (Appendix Fig. S1C,D). In *Dnmt3C^{KO/KO}* spermatocytes, we confirmed further transcriptional upregulation of L1 elements during meiosis, while ERVs remained consistently expressed (Fig. 1D). To compare the transcriptional responses of unmethylated ERVs and L1s, we plotted the average Z-score comparing the log(TPM) of each stage to all stages and the log(TPM) of expression values obtained from RNA sequencing in spermatogonia and spermatocytes (Fig. 1E). This further showed that L1s and ERVs are upregulated in *Dnmt3C^{KO/KO}* spermatogonia and while ERVK expression remains consistent, L1 expression is boosted at meiosis.

Taken together, our data show a dynamic expression of retrotransposons in the absence of local DNA methylation at their promoters following germ cell development. While ERVK expression in *Dnmt3C^{KO/KO}* mutants did not change substantially at the onset of meiosis, L1 transcription increased massively. This varied expression pattern indicates that the activating cues of unmethylated L1s and ERVKs differ in spermatogenesis.

## A switch from a bivalent to an active chromatin state might underpin transposon expression at meiosis

Previous studies in mESCs suggest that upon hypomethylation, retrotransposons are silenced by repressive histone modifications (Walter et al, 2016). Moreover, there is evidence that repressive H3K9me2 marks TEs in the absence of DNA methylation in *Dnmt3L* and *Miwi2* mutants spermatogonia (Di Giacomo et al, 2013; Zamudio et al, 2015). To investigate if the varied expression pattern of TEs we observed in *Dnmt3C* mutants correlates with specific dynamics of the chromatin landscape in spermatogenesis, we profiled key histone modifications in sorted germ cells. To this end, we performed CUT&Tag (Kaya-Okur et al, 2019) at three developmental stages: wild-type embryonic prospermatogonia

(E15.5) (Appendix Fig. S2A–C) as well as postnatal spermatogonia (Spg) and spermatocytes (Spc) in wild-type and *Dnmt3C^{KO/KO}* germ cells (Figs. 2A and EV2A). Specifically, we mapped two active (H3K4me3 and H3K27ac) and three repressive (H3K9me2, H3K9me3, and H3K27me3) histone modifications.

In prospermatogonia at E15.5, when TEs are transcribed at low level prior to piRNA-directed DNA methylation, we observed that DNMT3C targets were marked by both active H3K4me3, H3K27ac and repressive H3K27me3, H3K9me3 histone modifications, along with a weak signal of repressive H3K9me2 (Appendix Fig. S2B,C). In postnatal wild-type stages (Spg and Spc), where DNMT3C targets are stably repressed by DNA methylation (Fig. 1C,D,E), active histone marks are absent and only the repressive marks H3K9me3 and H3K9me2, were detected (Figs. 2A and EV2A). Conversely, in postnatal *Dnmt3C^{KO/KO}* Spg, H3K9me3 was significantly reduced, and H3K9me2 showed an enrichment at DNMT3C targets (Figs. 2A and EV2A). This aligns with previous findings indicating that H3K9me2 represses young retrotransposons at this stage when DNA methylation is deficient (Zamudio et al, 2015; Di Giacomo et al, 2014). Notably, we also observed strong retention of both active H3K4me3 and repressive H3K27me3 (Figs. 2A and EV2A), suggesting that a so called "bivalent signature" marks the promoters of young retrotransposons in the absence of DNA methylation. In addition, H3K27ac was retained at DNMT3C targets (Figs. 2A and EV2A), consistent with previous findings (Sakashita et al, 2020). Upon entry into meiosis, in *Dnmt3C^{KO/KO}* Spc, H3K27me3 became reduced along with an increase of the active H3K27ac mark (Figs. 2A and EV2A), suggesting that a transition from the bivalent towards an active chromatin signature takes place at some retrotransposon promoters.

It has previously been suggested that bivalent chromatin might play a crucial role in poising developmental gene expression (Bernstein et al, 2006; Boulard et al, 2015). In embryonic germ cells, H3K4me3-H3K27me3 bivalency at germline-specific gene promoters preserves these genes in a poised state, enabling their activation at the appropriate developmental stage of gametogenesis (Sachs et al, 2013). To test if the observed bivalency at TEs in *Dnmt3C^{KO/KO}* spermatogonia might similarly poise TEs for expression in meiosis, we dissected the occurrence of this bivalent mark at TEs throughout spermatogenesis. Peak calling on H3K4me3 and H3K27me3 CUT&Tag data from *Dnmt3C^{KO/KO}* spermatogonia allowed us to categorize DNMT3C targets into four chromatin signatures according to their occupancy: likely bivalent (overlapping peaks of H3K4me3 and H3K27me3), H3K4me3-marked, H3K27me3-marked, and uncategorized (no peaks detected)

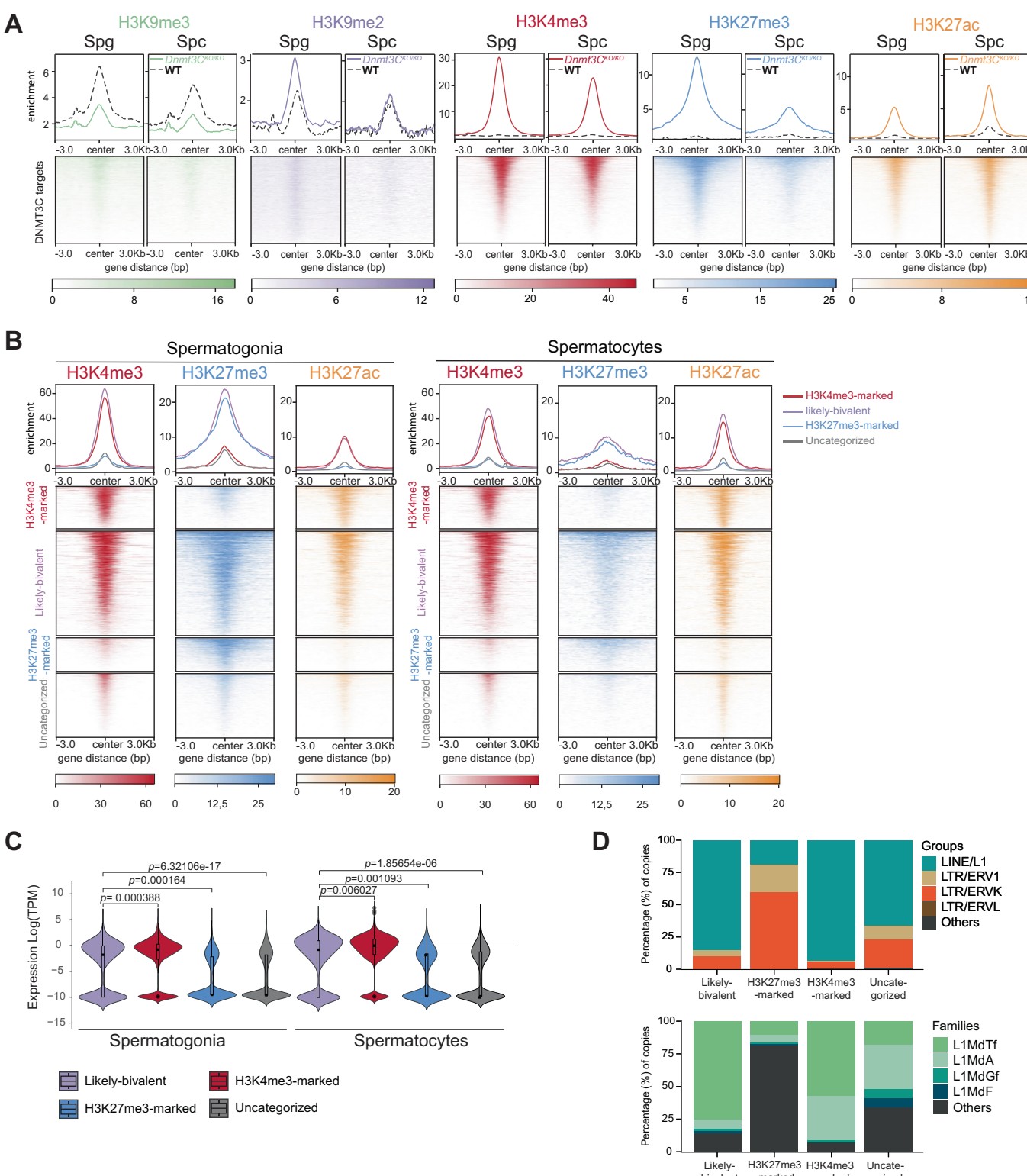

(Figs. 2B and EV2B; Appendix Figs. S2D,E and S3A–D). The bivalent H3K4me3-H3K27me3 mark was present across DNMT3C targets in all stages (Figs. 2B and EV2B; Appendix Fig. S2D,E). In *Dnmt3C*$^{KO/KO}$ Spg, we observed the highest level of H3K27me3, particularly in the likely bivalent and H3K27me3-marked

categories (Figs. 2B and EV2B; Appendix Fig. S2D). In *Dnmt3C*$^{KO/KO}$ Spc, we observed a widespread reduction in H3K27me3 across all categories and the re-emergence of active H3K27ac in H3K4me3-marked and likely bivalent DNMT3C targets (Figs. 2B and EV2B). These findings suggest that, upon

◄

**Figure 2.  A switch from a bivalent to an active state might underpin transposon expression at meiosis.**

(A) Heatmaps displaying normalized coverage and metaplots showing mean enrichment of H3K4me3, H3K27me3, H3K9me2, H3K9me3, and H3K27ac CUT&Tag, including multi-mapped reads merged from two biological replicates, centered on DNMT3C targets in sorted $Dnmt3C^{KO/KO}$ spermatogonia (Spg) and spermatocytes (Spc). The corresponding mean enrichment in wild-type is illustrated as a dashed black line in the same metaplot. (B) Heatmaps showing normalized coverage and metaplots displaying mean enrichment of H3K4me3, H3K27me3 and H3K27ac CUT&Tag, including multi-mapped reads centered on likely bivalent, H3K4me3-marked, H3K27me3-marked, and uncategorized DNMT3C targets in sorted $Dnmt3C^{KO/KO}$ Spg and Spc. (C) Violin plots with boxplots showing log(TPM) values from RNA sequencing of three biological replicates from TElocal in $Dnmt3C^{KO/KO}$ spermatogonia and spermatocytes of likely bivalent, H3K4me3-marked, H3K27me3-marked Uncategorized DNMT3C targets identified with multimapping reads. The centerline of the box plot indicates the median TPM, error bars represent 1.5 times the IQR above and below the median, where IQR is defined as the difference between the third and the first quartiles. All TEs with an average TPM count of 0 were plotted as −10. P values were tested using an unpaired t test between likely bivalent clusters and other clusters of the same stage and genotype. (D) Column charts showing the percentage of TE classes and families in each chromatin category of likely bivalent, H3K4me3-marked, H3K27me3-marked, and uncategorized DNMT3C targets from uniquely mapped reads of two biological replicates.

entry into meiosis, DNMT3C targets transition from a H3K4me3-H3K27me3 bivalent signature in spermatogonia, to an active state marked by H3K4me3 and H3K27ac.

Therefore, we next examined how this bivalent chromatin mark correlates with the transcriptional output of DNMT3C targets. To this end, we compared the TE expression observed in spermatogonia and spermatocytes across the four chromatin signatures identified in Spg. Indeed, in locally unmethylated $Dnmt3C^{KO/KO}$ mutants, retrotransposons classified as likely bivalent showed significantly higher expression levels than H3K27me3-marked or uncategorized DNMT3C targets but significantly lower levels than when H3K4me3-marked in both spermatogonia and spermatocytes (Figs. 2C and EV2C). To determine if this was correlated with different TE families, we compared the identified chromatin signatures in $Dnmt3C^{KO/KO}$ spermatogonia between L1 and ERV families. Among L1 retrotransposon families, the percentage of H3K4me3-marked copies increased progressively in the youngest families: L1MdT > L1MdA > L1MdGf > L1MdF (Figs. 2D and EV2D; Appendix Fig. S4A). Hence, there is a direct correlation between retrotransposition potential and the presence of active chromatin marks at the promoters of unmethylated younger L1 families. Interestingly, the likely bivalent category was composed predominantly of L1MdT elements, whereas other categories included a variety of TE families (Fig. 2D). ERVs were mostly categorized as H3K27me3-marked, while ERVKs were the most abundant ERV subclass in the H3K4me3-marked category (Figs. 2D and EV2D), consistent with IAP and MMERVK10C being among the most upregulated ERV types in $Dnmt3C^{KO/KO}$ (Fig. 1C,D).

To determine whether chromatin signatures influence TE expression in a family-dependent manner, we compared the expression of different subfamilies across chromatin signatures in spermatogonia and spermatocytes (Appendix Fig. S4B). Interestingly, not all LINE1 subfamilies showed higher expression when marked by H3K4me3 compared to likely bivalent-marked, and H3K27me3-marked LINE1 copies exhibited only modestly reduced expression in both cell types (Appendix Fig. S4B). This suggests that H3K27me3 does not strongly repress LINE1s. In contrast, ERVK elements displayed a clear expression difference: H3K4me3-marked copies were more highly expressed than those marked by likely bivalent or H3K27me3 in both spermatogonia and spermatocytes, indicating effective repression of ERVKs by H3K27me3 (Appendix Fig. S4B).

Overall, our data demonstrates that in the absence of DNA methylation, likely bivalent and H3K4me3-marked DNMT3C targets are more prone to transcriptional reactivation compared

to H3K27me3-marked or uncategorized TEs. This suggests that active chromatin marks may play a role in regulating retrotransposon expression, highlighting the dynamic interplay between chromatin states and TE activity in $Dnmt3C$-deficient germ cells. Moreover, our data indicate that the majority of the likely bivalent TEs belong to the L1MdT type and ERVKs are more effectively repressed by H3K27me3 than L1s. The variable expression of TE families observed across different chromatin signatures suggests that additional regulatory cues may modulate TE expression in the absence of DNA methylation.

## Identifying candidate regulators of retrotransposons in male germ cells

We sought to identify DNA methylation-dependent binding proteins responsible for retrotransposons' transcription. Previous work in cell culture models suggested two possible models for achieving repression by DNA methylation at TEs, the first is the recruitment of a repressive complex via TFs that bind to their motifs only when cytosines are methylated (Boulard et al, 2020), the other model posits that DNA methylation inhibits the binding of activating TFs (Kaluscha et al, 2022). Because the identity of the relevant TFs in germ cells is unknown, the mechanism of TE silencing by DNA methylation remains elusive.

To discover the TFs, present in germ cells that bind to retrotransposon promoters in a DNA methylation-dependent manner, we conducted an unbiased proteomic screen consisting of biotin/streptavidin-based DNA bait pulldown experiments using nuclear protein extracts from mouse germ cells, followed by protein detection via mass spectrometry. We used DNA baits representative of retrotransposon promoters that reactivate in $Dnmt3C^{KO/KO}$, hence normally silenced by DNA methylation. Specifically, we used a consensus IAP LTR1a promoter and L1MdT promoter monomers (repeats 1 and 2) (Fig. EV3A,B). As bait controls, we designed and synthesized scrambled DNA sequences matching the TE promoter sequences in length, nucleotide composition, and CpG positions (Fig. EV3A,B). Half of the baits were methylated in vitro at CpG sites, while the other half remained unmethylated, with methylation status confirmed by bisulfite cloning (Fig. EV3C,D) (Table EV1). Among the proteins consistently detected in pulldowns with unmethylated baits, nuclear respiratory factor 1 (NRF1) was enriched as a sequence-specific binder of both IAP and L1 baits (Fig. 3A,B; Appendix Figs. S5 and S6). NRF1 was previously shown to be a DNA methylation-sensitive TF that is outcompeted at most of its motifs when the CpG sites are methylated (Domcke et al,

2015). Furthermore, NRF1 plays a key role in spermatogenesis by regulating germline-specific genes (Wang et al, 2017; Wang et al, 2024). Another DNA methylation-sensitive TF uncovered in the unmethylated IAP pulldowns was CREB1, previously shown to bind IAPs in DNA methylation-deficient neurons (Kaluscha et al, 2022). However, CREB1 was not enriched in a sequence-specific manner (Appendix Fig. S5).

We performed in silico Analysis of Motif Enrichment (AME) (McLeay and Bailey, 2010) to identify enriched TF motifs within the IAP and L1 fractions of DNMT3C targets, using all ERVK and LINE1 promoters as a background. NRF1 and CREB1 emerged as top hits, being enriched above background and also expressed in germ cells (Dataset EV2). We then overlapped the TFs identified in silico with our pulldown-MS data. At IAPs, NRF1, NR0B1, and CREB1 motifs were found to be enriched as well as expressed in germ cells (Fig. 3C). Similarly, NRF1 and RBPJ were expressed and identified at LINE1s (Fig. 3C). Among these four TF candidates, we further focused on CREB1 and NRF1, both of which have previously been shown to be DNA methylation-sensitive TFs (Domcke et al, 2015; Kaluscha et al, 2022) and are known to bind and regulate genes in germ cells (Martianov et al, 2010; Wang et al, 2017).

To test whether CREB1 and NRF1 could bind differentially to retrotransposon promoters in a DNA methylation-dependent fashion, we performed CUT&Tag for both TFs in Spg from wild-type and $Dnmt3C^{KO/KO}$ mice. CUT&Tag for CREB1 showed an overall weak signal, even at known target genes (Friedrich et al, 2020; Fig. EV3E), and no noticeable differences were observed at retrotransposon promoters between wild-type and $Dnmt3C^{KO/KO}$ (Fig. EV3F). In contrast, CUT&Tag for NRF1 showed enrichment at previously identified targets (Wang et al, 2017; Fig. EV3E) and a slightly higher occupancy at DNMT3C targets in $Dnmt3C^{KO/KO}$ compared to wild-type germ cells (Fig. EV3F). We therefore focused on NRF1 as a potential candidate methylation-sensitive trans-activator of unmethylated young retrotransposons.

## The DNA methylation-sensitive transcription factor NRF1 binds unmethylated transposon promoters

After confirming NRF1 expression and nuclear localization in E15.5 prospermatogonia, postnatal spermatogonia, and spermatocytes (Fig. EV4A), we used CUT&Tag to profile NRF1 in all three germ cell stages purified from wild-type and $Dnmt3C^{KO/KO}$. While NRF1 occupancy at DNMT3C targets was barely detectable in wild-type prospermatogonia, its binding was significantly increased in $Dnmt3C^{KO/KO}$ spermatogonia and spermatocytes (Figs. 4A–C and EV4B). We also observed a higher NRF1 occupancy at certain unmethylated LINE1 and LTR copies upon inspecting CUT&Tag tracks (Fig. EV4C,D). Peak calling on NRF1 CUT&Tag data revealed a noticeable overlap of NRF1 peaks with DNMT3C targets, increasing from approximately 3.5% in $Dnmt3C^{KO/KO}$ spermatogonia to 9% overlap in $Dnmt3C^{KO/KO}$ spermatocytes (Figs. 4D and EV4E; Appendix Fig. S7A. Determining differential enrichment of NRF1 by RepEnrich analysis followed by comparison with EdgeR revealed that NRF1 binding was significantly enriched at several retrotransposon copies in $Dnmt3C^{KO/KO}$ spermatogonia and spermatocytes compared to wild-type (Fig. 4B,C). Moreover, the binding of NRF1 to certain TEs in $Dnmt3C^{KO/KO}$ spermatogonia and spermatocytes was significantly enriched over IgG controls, while

in wild-type conditions no differences were observed (Appendix Fig. S7B,C). Interestingly, NRF1 binding was specifically upregulated at L1 and IAP subfamilies that also reactivate in $Dnmt3C^{KO/KO}$ indicating that NRF1 binding might activate these TEs (Figs. 4E and 1C,D; Appendix Fig. S7D,E). Our results indicate that NRF1 binds to unmethylated promoters of evolutionarily young retrotransposons in spermatogenesis.

## NFR1 is required for the expression of unmethylated IAPs in the male germline

To investigate the possible causal role for NRF1 in activating the transcription of retrotransposons in the absence of DNA methylation, we generated a conditional $Nrf1$ germ cell-specific knockout ($Nrf1^{cKO/KO}$) in the $Dnmt3C^{KO/KO}$ and wild-type backgrounds. To this end, we crossed mice bearing a floxed allele of $Nrf1$ ($Nrf1^{fl/fl}$) (Wang et al, 2017) with Ddx4-Cre (Vasa-Cre) driver mice, which would induce a conditional knockout in germ cells by E18.5 (Gallardo et al, 2007), and bred them back into the $Dnmt3C^{KO/KO}$ background (Barau et al, 2016) (Appendix Fig. S8A). Consistent with previous studies on the germ cell-specific ablation of $Nrf1$ with the $Ddx4$-Cre driver (Wang et al, 2017), we observed loss of germ cells in $Nrf1^{cKO/KO}$ males, becoming apparent by P8 and more pronounced by P10 (Fig. EV5A,B; Appendix Fig. S8B). To minimize any confounding effects due to germ cell loss in the $Nrf1^{cKO/KO}$ background, we focused on collecting testes from conditional mutants and controls at P5, a developmental stage where spermatogonia and spermatogonial stem cells are the sole germ cell populations (Fig. EV1A). This enabled us to assess the effects of NRF1 depletion on TE transcription in spermatogonia, prior to extensive germ cell loss. For all downstream analysis, we sampled total RNA, total protein, and fixed tissue for immunofluorescence and tested genotypes from the same animals with matching littermate controls whenever possible. Following PCR verification of the compound genotypes in the collected samples (Appendix Fig. S8C) (Table EV1), we confirmed the CRE-mediated conversion into the knockout allele of $Nrf1$ in $Vasa^{Cre}$ testes by the absence of NRF1 immunofluorescence detection in germ cells (Fig. 5A). As expected, nuclear NRF1 was present in wild-type and $Dnmt3C^{KO/KO}$ germ cells but consistently absent in $Nrf1^{cKO/KO}$ and double knockout (dKO) of $Nrf1^{cKO/KO}$ and $Dnmt3C^{KO/KO}$ germ cells, confirming successful loxP recombination and the efficient knockout of $Nrf1$ (Fig. 5A).

To assess the impact of $Nrf1$ knockout on retrotransposon expression, we performed poly-A-enriched RNA sequencing on whole testes at P5 and analyzed differentially expressed TE families through pairwise comparisons of the mutants. Expectedly, the comparison between the transcriptome of $Dnmt3C^{KO/KO}$ and wild-type controls showed significant upregulation of three of the youngest ERVK subfamilies: IAPEz-int, IAPA_Mm-int and MMERVK10C-int alongside their most commonly associated LTRs (Fig. 5B). When compared to our RNA sequencing from sorted spermatogonia (Fig. 1C), the absence of upregulation of any of the young LINE1 subfamilies in $Dnmt3C^{KO/KO}$ stood out as a notable difference. This difference is likely due to background RNA derived from whole-testis samples rather than sorted cells, consistent with previous whole-testis RNA-sequencing analysis of $Dnmt3L^{KO/KO}$ showing upregulation of only ERVK subfamilies at P10 (Zamudio et al, 2015). The comparison between $Nrf1^{cKO/KO}$ and wild-type showed a trend towards downregulation of most young

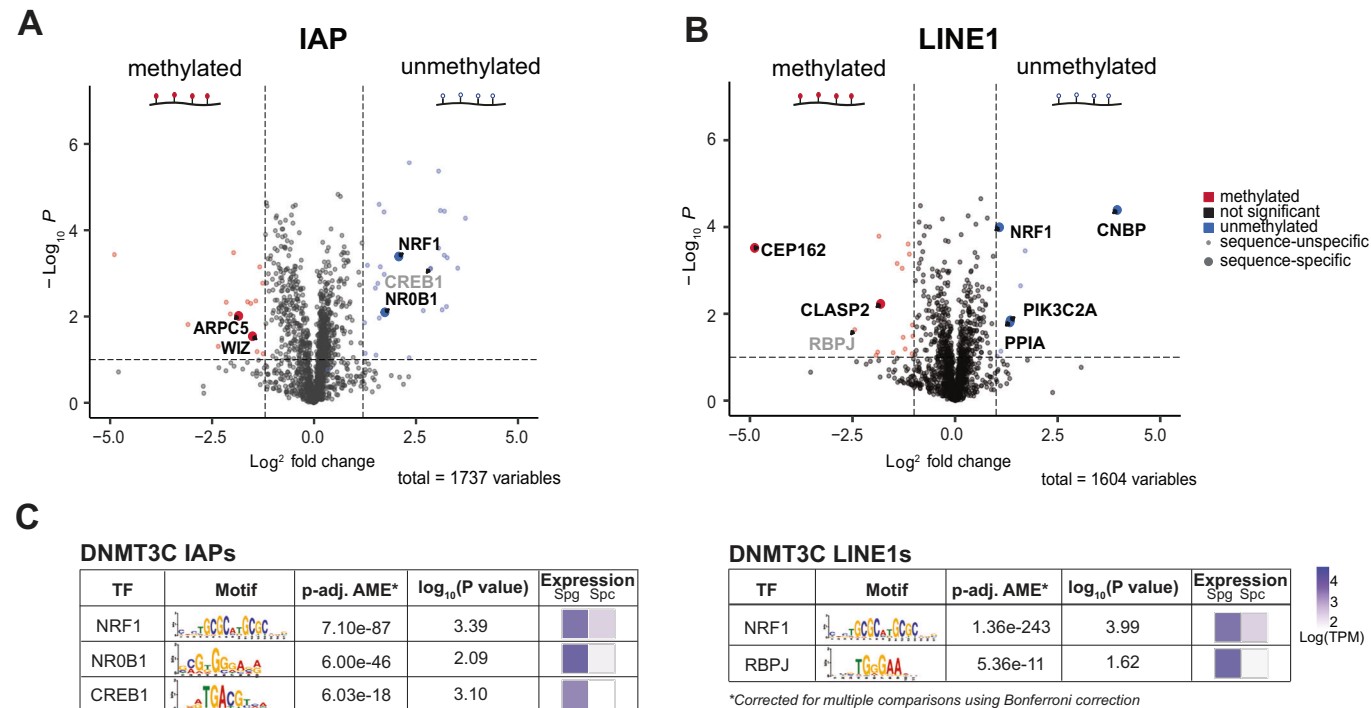

**Figure 3. Identifying candidate regulators of retrotransposons in male germ cells.**

(A) Volcano plot showing -log₁₀P values from Welsh t test and log₂ FC of proteins enriched on methylated and unmethylated IAP LTR1a from three biological replicates detected by mass spectrometry in DNA-pulldown experiments. Sequence-unspecific proteins that were not detected as enriched on the scrambled controls are represented as small points, while sequence-specific proteins are highlighted by larger, filled points. Labeled proteins refer to sequence-specifically enriched proteins apart from CREB1 and RBPJ labeled in gray. (B) As in (A) for methylated and unmethylated LINE1MdT. (C) Table of TFs detected in (A) alongside AME motif enrichment on the sequence of the bait (IAP LTR1a and L1MdT) using all transposons (all IAPs and all LINE1s) as background. Transcriptional expression in Spg and Spc is illustrated as a heatmap for log(TPM).

retrotransposons, with only IAP1_Mm-int above the significance cutoff value (Fig. EV5C).

In contrast, several protein-coding genes were differentially expressed between *Nrf1^cKO/KO* and wild-type (Dataset EV3), with fewer upregulated (1014) than downregulated (1828), including established germline gene targets of NRF1 (Wang et al, 2017). This supports NRF1's function as a TF regulating germline gene expression. Importantly, our differential TE expression analysis in *Nrf1^cKO/KO* testes indicates that the germ cell loss in *Nrf1* conditional mutants does not compromise the reliability of TE expression analysis using whole-testis RNA sequencing at P5.

To better observe the impact of the germline-specific conditional knockout of *Nrf1* on the transcription of unmethylated TEs, we made a series of pairwise comparisons using the levels of TE upregulation in *Dnmt3C^KO/KO* over wild-type as a reference (Fig. 5B). When comparing double "dKO" (*Nrf1^cKO/KO*; *Dnmt3C^KO/KO*) vs. *Nrf1^cKO/KO* (Fig. 5C) to the reference comparison (Fig. 5B), the dKO showed overall reduced upregulation of several IAPs and their associated LTRs (Fig. 5C). Specifically, IAPA_MM-int, IAPLTR2a, IAPLTR2_Mm, IAPLTR1_Mm, and IAPEz-int subfamilies were considerably less upregulated in dKO comparisons than in reference comparisons of *Dnmt3C^KO/KO* testes (Fig. 5D). Remarkably, these IAP types were also among the subfamilies showing the strongest differential NRF1 binding between Dnmt3CKO/KO and wild-type germ cells (Fig. 4E). This result demonstrates that NRF1

is required for the full-blown transcription of IAP in unmethylated spermatogonia.

Even though we could not observe LINE1 upregulation in *Dnmt3C^KO/KO* alone when compared to wild-type (Fig. 5B), L1Md_A was found to be upregulated in dKO, but not in single *Nrf1^cKO/KO* (Figs. 5C and EV5C). This suggests that an additive effect between the loss of DNA methylation and the loss of NRF1 in germ cells might lead to increased L1Md_A transcription. When comparing dKO to wild-type or to *Dnmt3C^KO/KO*, IAP expression was even further reduced, though it remains possible that some extent of the differences in these pairwise comparisons could be attributed to differential germ cell loss (Figs. 5D and EV5D–F). In contrast, MMERVK10C-int and their most commonly associated LTRs, RLTR10C, RLTR10B and RLTR10B2, maintained the same expression levels in dKO as in *Dnmt3C^KO/KO* (Figs. 5D and EV5D). Hence, these elements do not require NRF1 for their activation.

Next, we evaluated TE expression at the protein level by immunofluorescence stainings of LINE1-ORF1 and IAP-POL in testis cryosections (Fig. 5E,F). As expected, LINE1-ORF1 was undetectable in *Nrf1^cKO/KO* and wild-type germ cells but was present in *Dnmt3C^KO/KO* testes. In agreement with our transcriptomic data on dKO mutants, LINE1-ORF1 expression in dKO germ cells was similar to that in *Dnmt3C^KO/KO* (Fig. 5E; Appendix Fig. S8D). We also detected IAP-POL expression in germ cells of *Dnmt3C^KO/KO* testis sections, but not in wild-type or *Nrf1^cKO/KO*, which showed

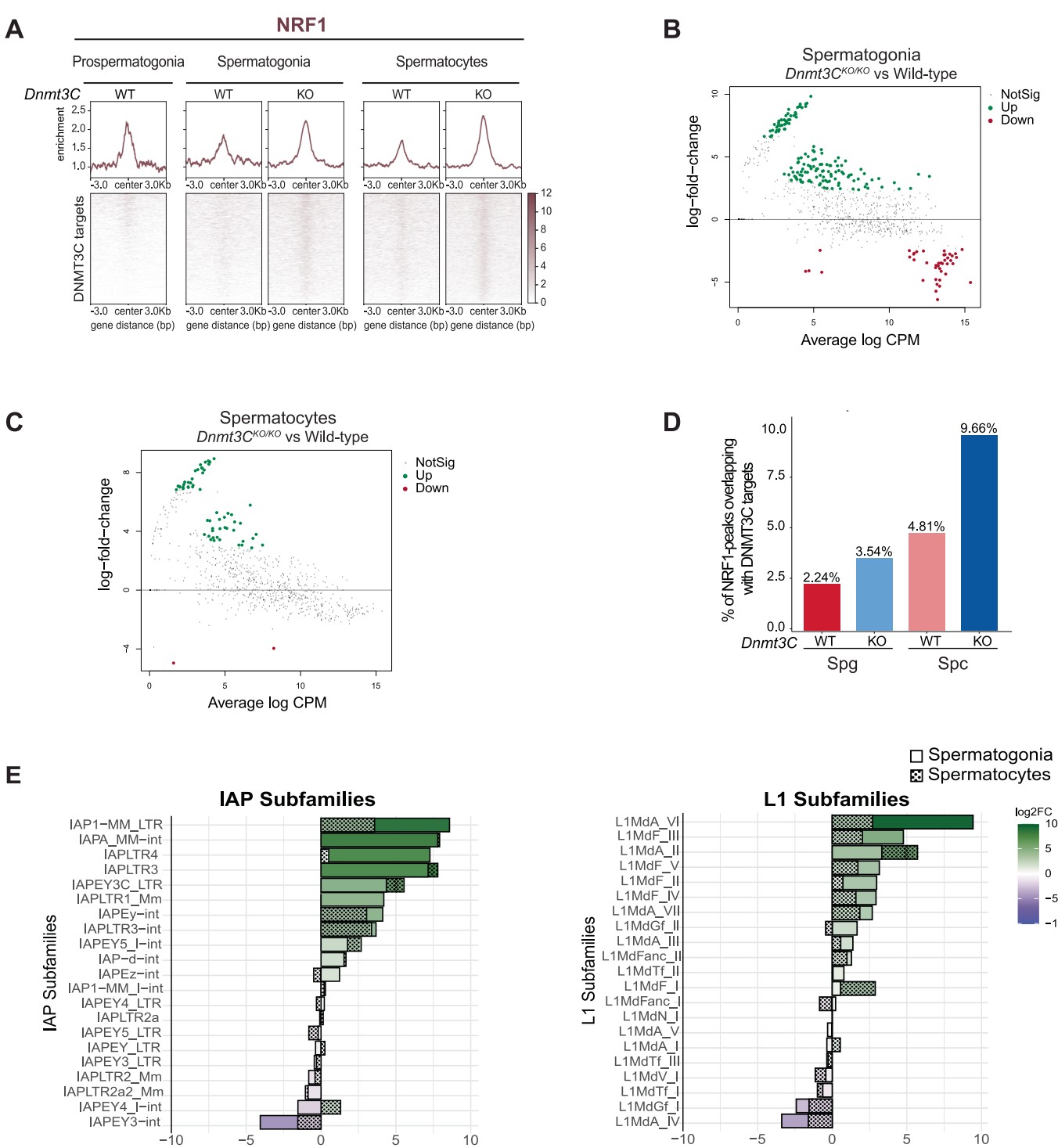

**Figure 4. The DNA methylation-sensitive transcription factor NRF1 binds unmethylated transposon promoters.**

(A) Heatmaps displaying normalized coverage and metaplots showing mean enrichment of NRF1 CUT&Tag, centered on DNMT3C targets from two biological replicates of sorted wild-type embryonic germ cells at E15.5, wild-type and $Dnmt3C^{KO/KO}$ sorted Spg and Spc. (B) MA plot showing differential NRF1 enrichment in $Dnmt3C^{KO/KO}$ vs. WT Spg, plotting log$_2$ fold change (log$_2$FC) against average log counts per million (logCPM) from two biological replicates. TE read counts were quantified using RepEnrich2. Significantly upregulated (Up, red), downregulated (Down, blue), and non-significant (NotSig, black) DNMT3C targets identified using edgeR. (C) As in (B) for Spc. (D) Bar chart showing percentage of DNMT3C targets overlapping with NRF1 peaks in WT and $Dnmt3C^{KO/KO}$ Spg and Spc. (E) Bar plots showing log$_2$FC NRF1 enrichment in $Dnmt3C^{KO/KO}$ versus wild-type for differentially enriched IAP (left) and L1 (right) subfamilies in Spg and Spc.

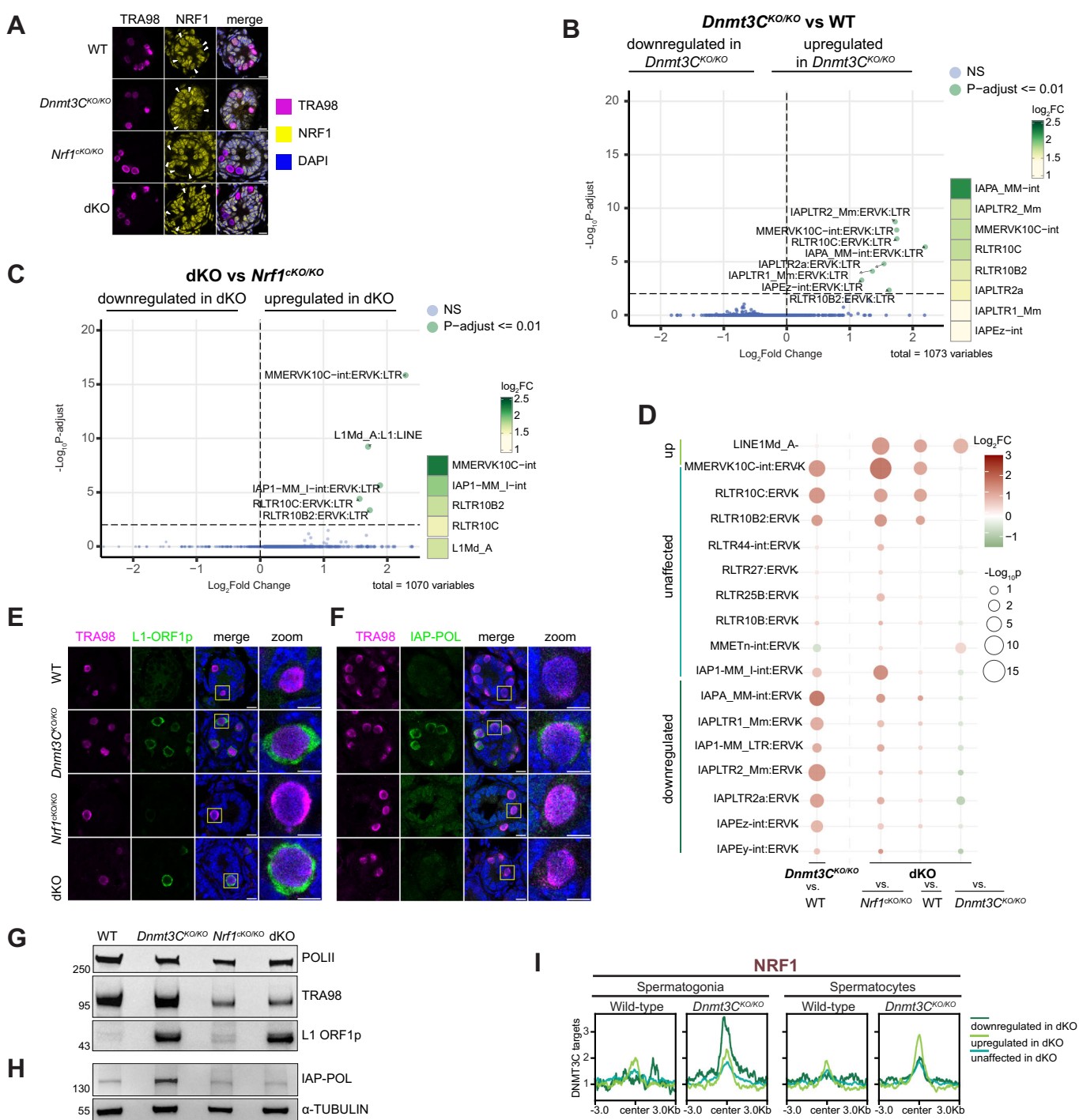

only background staining. Remarkably, in the germ cells of dKO sections, IAP-POL staining levels were comparable to the background level observed in wild-type and $Nrf1^{cKO/KO}$. This indicates that despite the loss of DNA methylation, the depletion of NRF1 leads to a nearly complete rescue of the IAP protein silencing (Fig. 5F).

Western blot analysis of LINE1-ORF1 confirmed our previous findings. Despite a reduction of TRA98 levels at P5 in dKO and $Nrf1^{cKO/KO}$, suggesting germ cell loss, LINE1-ORF1 expression

remained similar to $Dnmt3C^{KO/KO}$ (Fig. 5G). By P8, LINE1-ORF1 levels appeared reduced in dKO compared to $Dnmt3C^{KO/KO}$, likely due to the progressive germ cell loss, which was evident from a decreased TRA98 signal (Fig. EV5G). Western blot analysis also revealed a specific IAP-POL band in $Dnmt3C^{KO/KO}$ extracts, which was remarkably reduced in dKO extracts to a similar level as in wild-type and $Nrf1^{cKO/KO}$ (Fig. 5H). These results align with our findings, indicating that while unmethylated LINE1s remain upregulated, silencing of IAP elements is rescued in dKO testes

**Figure 5. NFR1 is required for the expression of unmethylated IAPs in the male germline.**

(A) Representative immunostaining of NRF1, TRA98 (germ cell marker) and DAPI (nuclear staining) on cryosections of wild-type, $Dnmt3C^{KO/KO}$, $Nrf1^{cKO/KO}$ and dKO ($Dnmt3C^{KO/KO}$, $Nrf1^{cKO/KO}$) testes at P5. Yellow arrowheads indicate germ cells, Scale bar, 10μm. The staining was performed in three biological replicates. (B) Volcano plot showing DESeq2 results for TEs With -$\log_{10}$P-adjusted (Wald test) and $\log_2$ FC values from three biological replicates in $Dnmt3C^{KO/KO}$ vs. wild-type displayed as TEname:family:TEclass. Non-significant (NS) data points depicted in blue, significant values in green. The heatmap displays $\log_2$ fold-change values for the same comparison. (C) As in (B) for dKO vs. $Nrf1^{cKO/KO}$. (D) Bubble chart showing DESeq2 results with $\log_2$ FC as a heatmap and $-\log_{10}$ $P$-adjusted values as circle size, for TE subfamilies (TEname:family:TEclass) differentially enriched in pairwise mutant comparisons from three biological replicates. The pairwise comparisons were analyzed for $Dnmt3C^{KO/KO}$ vs. wild-type and dKO vs. $Nrf1^{cKO/KO}$, wild-type and $Dnmt3C^{KO/KO}$. (E) As in (A) showing a representative immunostaining of LINE1-ORF1p and TRA98 from two biological replicates. (F) As in (E) of IAP-POL and TRA98. Scale bar, 10 μm (5 μm in cropped images). (G) Representative Western blot analysis of POL II, TRA98 and LINE1-ORF1p on protein extracts from wild-type, $Dnmt3C^{KO/KO}$, $Nrf1^{KO/KO}$ and dKO testes at P5, performed in two biological replicates. (H) As in (G) for a-TUBULIN and IAP-POL. L1-ORF1p and IAP-POL blots were developed using femto chemistry. (I) Metaplots showing mean enrichment of NRF1 CUT&Tag in wild-type and $Dnmt3C^{KO/KO}$ Spg and Spc, centered on DNMT3C targets that were upregulated, downregulated or unaffected in dKO vs. $Nrf1^{cKO/KO}$ compared to $Dnmt3C^{KO/KO}$ vs. wild-type. The signal was merged from two biological replicates. Source data are available online for this figure.

at P5. Altogether, our functional data revealed that NRF1 is required for the expression of unmethylated IAPs in male premeiotic germ cells.

To investigate whether the downregulated DNMT3C targets in dKO testes at P5 have distinct NRF1 binding patterns, we examined NRF1 binding at DNMT3C targets categorized into downregulated, upregulated, and unaffected in dKO vs. $Nrf1^{cKO/KO}$ compared to $Dnmt3C^{KO/KO}$ vs. wild-type (Fig. 5). In $Dnmt3C^{KO/KO}$ spermatogonia, NRF1 was more enriched at downregulated DNMT3C targets than at other targets. This pattern was reversed in $Dnmt3C^{KO/KO}$ spermatocytes, where NRF1 binding increased at DNMT3C targets, which were upregulated in dKO at P5, suggesting that these TEs, specifically LINE1s, might only be affected by NRF1 regulation at the onset of meiosis (Fig. 5I). Additionally, plotting H3K4me3 and H3K27me3 at the same categories of DNMT3C targets, revealed that TEs downregulated in dKO at P5 exhibit distinct chromatin signatures (Fig. EV5H). In $Dnmt3C^{KO/KO}$ spermatogonia, down-regulated DNMT3C targets were not marked by H3K27me3, showing only enrichment for H3K4me3, whereas the other DNMT3C targets showed enrichment for both H3K4me3 and H3K27me3. By contrast, in $Dnmt3C^{KO/KO}$ spermatocytes, H3K27me3 was lost at DNMT3C targets, coinciding with increased NRF1 binding (Fig. EV5H), suggesting a potential link between NRF1 binding and chromatin modifications. These findings uncover NRF1 as the trans-activator of H3K4me3-marked IAPs responsible for their expression in male premeiotic germ cells when DNA methylation is compromised (Fig. 6).

## Discussion

In this work, we used the $Dnmt3C^{KO/KO}$ mouse model to gain insights into how retrotransposons leverage histone modifications and transcription factors when DNA methylation at TE promoters is defective. Previous work established that DNA methylation can be largely dispensable for LINE1 silencing before the onset of meiosis due to their targeting by repressive H3K9me2 in spermatogonia (Di Giacomo et al, 2014; Zamudio et al, 2015; Di Giacomo et al, 2013). By combining expression analysis of TE-encoded proteins in spermatogenesis and RNA sequencing of purified premeiotic spermatogonia and meiotic spermatocytes, we also uncovered a role of DNA methylation in repressing LINE1 in spermatogonia (Fig. 1). We demonstrate that loss of promoter DNA methylation impacts LINE1 and ERVK families differently: in

$Dnmt3C$ mutant germ cells, ERVKs remain active throughout spermatogenesis, whereas LINE1s show low expression before meiosis with a marked increase in activity at meiosis.

We further show that the resulting patterns of TE expression correlated with specific and dynamic combinations of histone modifications. Nearly all unmethylated TE promoters in $Dnmt3C^{KO/KO}$ mutants retained active H3K4me3 and gained H3K27me3 in spermatogonia, bearing a bivalent chromatin signature that could poise TEs for reactivation at a later developmental stage. We observed that TE promoters marked by high levels of H3K27me3 in spermatogonia showed a sharp decrease of this mark in spermatocytes (Fig. 2A,B) correlating with higher transcriptional output in meiosis (Fig. 1). Interestingly, loss of H3K27me3 is not observed in the strongly hypomethylated oocytes, where TEs are effectively silenced by this mark (Huang et al, 2021). Bivalency has been reported to protect repressed genes from irreversible silencing by aberrant DNA methylation in cancer (Bernstein et al, 2006; Boulard et al, 2015). There is ample evidence of low-level retrotransposition in the mouse germline (Gagnier et al, 2019), indicating that a small number of TE copies can occasionally escape DNA methylation-dependent silencing. How TEs achieve this in wild-type germ cells is unknown. However, our findings from $Dnmt3C$ mutant germ cells suggest that the acquisition of a bivalent chromatin at TE promoters may enable these elements to evade silencing by the default, untargeted de novo DNA methylation happening at the rest of the genome. On the other hand, the marking of TE promoters with H3K27me3 in the absence of DNA methylation may also reflect a fallback state to an ancient pathway of TE silencing that is conserved across many eukaryotes (Hure et al, 2024; Miró-Pina et al, 2022; Gahan et al, 2024; Huang et al, 2021). Finally, our data suggests that the bivalent mark does not repress all TE families uniformly—only few L1MdAs were likely bivalently marked (Fig. 2D; Appendix Fig. S4B). This implies that an additional regulatory layer such as the H3K9me2 mark (Di Giacomo et al, 2014) may be the main determinant of their transcriptional output in spermatogonia.

Data from our in vivo experiments reinforce the current model that the direct inhibition of TF binding is the prevailing mode of TE repression by DNA methylation (Kaluscha et al, 2022). In agreement with Kalusha et al, we did not detect any methyl binding protein (MBD) repressors in our pulldowns using in vitro methylated TE promoters (Fig. 3A,B; Appendix Figs. S5 and S6). Instead, we captured methyl-sensitive TFs, whose binding is inhibited by DNA methylation (Fig. 3A,B). This suggests that also

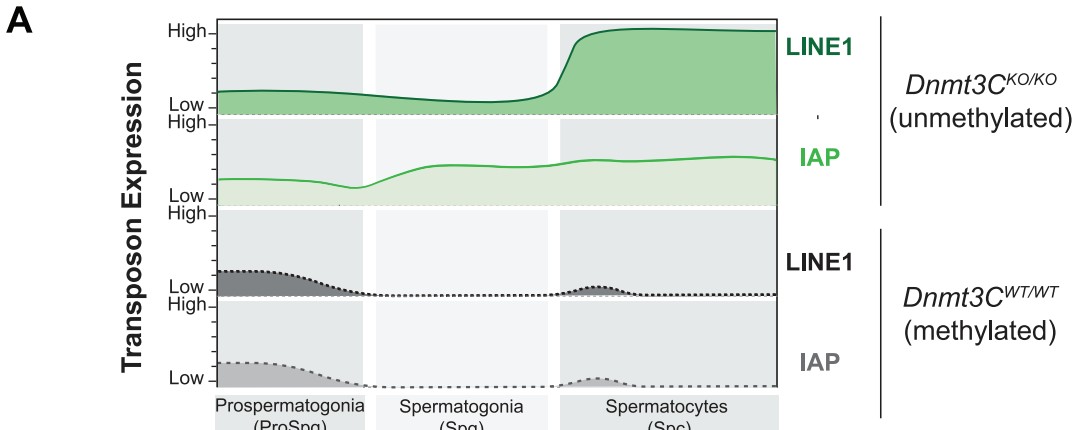

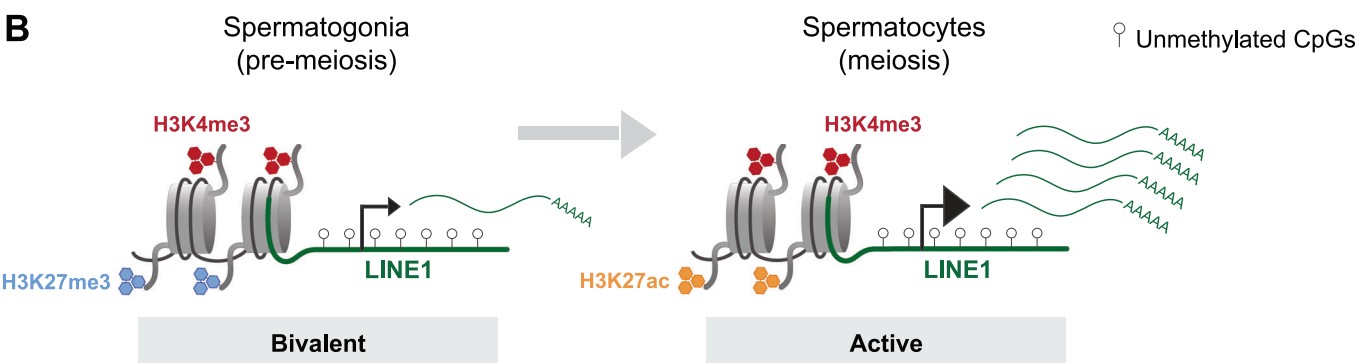

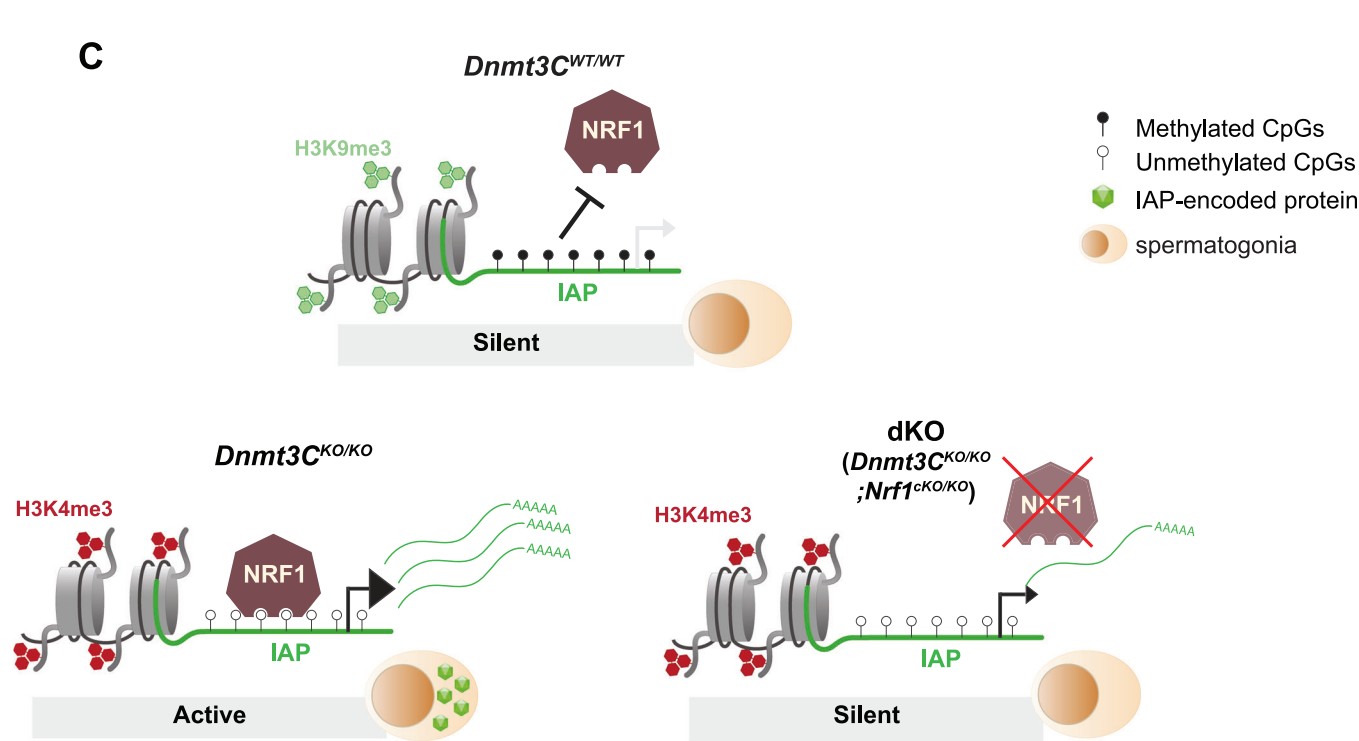

◀ **Figure 6. Model for retrotransposon regulation in mice germ cells.**

(A) Schematic representation of LINE1 and IAP expression dynamics in *Dnmt3C*$^{KO/KO}$ (green) and *Dnmt3C*$^{WT/WT}$ (gray) throughout spermatogenesis. (B) A chromatin switch from bivalent H3K27me3-H3K4me3 (spermatogonia) to active H3K27ac-H3K4me3 (spermatocytes). Unmethylated LINE1 elements switch from a bivalent state (low transcriptional, poised for reactivation in meiosis) to an active state in spermatocytes (high transcriptional output). (C) Model of IAP regulation in spermatogonia. In *Dnmt3C*$^{WT/WT}$, IAPs are silenced by DNA methylation and H3K9me3; Low/No NRF1 occupancy; No IAP transcription and translation. In *Dnmt3C*$^{KO/KO}$, IAPs are activated by the absence of DNA methylation and presence of H3K4me3; Higher NRF1 occupancy; High IAP transcription and translation. In dKO (*Dnmt3C*$^{KO/KO}$; *Nrf1*$^{cKO/KO}$), unmethylated IAPs are silenced by the loss of NRF1 trans-activator; Low/No IAP transcription no IAP translation.

in germ cells TE repression by DNA methylation is mediated through inhibition of TF binding. However, it is important to note that the silencing mechanism of methylated retrotransposons likely differs between germ cells and somatic cells, which require the scaffolding protein TRIM28 (also known as KAP-1) (Rowe et al, 2010; Boulard et al, 2020).

Our study identifies NRF1 as a DNA methylation-sensitive TF required for the transcription of hypomethylated IAPs in spermatogonia. Using *Nrf1* germline conditional alleles in the *Dnmt3C*$^{KO/KO}$ background, we found that transcription of IAPs is reduced, and the silencing of the IAP-encoded protein POL (reverse transcriptase) is rescued to the levels observed in wild-type spermatogonia (Figs. 5C,D,F,H and EV5D). To our knowledge, this is the first evidence of a TF that promotes transcription of unmethylated IAPs in germ cells. We also identified CREB1 in our pulldown experiments, but failed to show its binding in germ cells (Figs. 3A and EV3E,F) despite the fact that CREB1 has been described to promote IAP transcription in neuronal cell culture models with defective DNA methylation (Kaluscha et al, 2022). However, CUT&Tag was found to be ineffective for the chromatin profiling of many TFs, and further technical limitations attributed to low-input material from primary tissues might impact the sensitivity of our experiments when profiling NRF1 or in particular CREB1 (Kaya-Okur et al, 2020). Thus, we cannot exclude that our failure to detect CREB1 binding to unmethylated IAPs in germ cells is due to some of the technical limitations of CUT&Tag. Our genomic profiling of NRF1 showed a significant increase over some retrotransposons in *Dnmt3C* mutants compared to wild-type in both spermatogonia and spermatocytes (Fig. 4B,C). While some studies have suggested potential global DNA demethylation events during meiosis (Gaysinskaya et al, 2018; Huang et al, 2023), we observed that NRF1 binding increases at DNMT3C target transposons during this stage despite reduced NRF1 expression, arguing against a global redistribution of NRF1 binding (Figs. 4A,E and 5I). Although we identified NRF1 binding to L1s in *Dnmt3C*$^{KO/KO}$, suggesting its role as a transcriptional regulator of L1s, we did not observe any reduction of L1-ORF1 protein. Paradoxically, L1MdA transcription even increased in dKO (*Nrf1*$^{cKO/KO}$; *Dnmt3C*$^{KO/KO}$) testes (Figs. 5C,D and EV5D). This highlights some of the limitations of our approach: NRF1 is also an essential TF for spermatogenesis, whose absence leads to progressive germ cell loss and impacts the transcription of important genes. For example, L1MdA activity in dKO can perhaps be explained by a down-regulation of EHMT1, which is a known NRF1 target (Di Giacomo et al, 2014; Zamudio et al, 2015; Palmer et al, 2019). EHMT1 is a histone methyltransferase that catalyzes the deposition of repressive H3K9me2, previously associated with L1 silencing in spermatogonia (Di Giacomo et al, 2014; Zamudio et al, 2015; Palmer et al, 2019). We found that, unlike L1MdTs, L1MdAs are not predominantly silenced by likely bivalent or H3K27me3 marks and might instead primarily rely on H3K9me2-mediated silencing in the absence of DNA methylation (Fig. 2A,D). Thus, reduction of H3K9me2 upon depletion of NRF1 might lead to the premature upregulation of L1MdA in spermatogonia lacking DNA methylation.

Finally, our data show that NRF1 is enriched at some unmethylated L1 subfamilies in spermatogonia, with even stronger enrichment observed in spermatocytes (Fig. 4D,E). This suggests that certain L1 subfamilies may also depend on NRF1 for transcription already in spermatogonia but potentially depend even more on NRF1 at the onset of meiosis, when L1 expression is boosted, repressive marks are lost and NRF1 binding is enhanced. However, we could not directly test this effect at meiosis in our *Nrf1* conditional mutants due to severe premeiotic spermatogenic arrest and germ cell loss caused by *Vasa-Cre*-driven conditional *Nrf1* ablation (Fig. EV5A,B; Appendix Fig. S8B). In spermatogonia, we observed an unexpected transcriptional upregulation of L1MdA in dKO testes compared to *Dnmt3C*$^{KO/KO}$ (Fig. 5D). However, when addressing this by immunofluorescence, it revealed equivalent L1-ORF1 levels in germ cells of both genotypes (Appendix Fig. S8D), indicating a disparity between transcript abundance and protein output. This discrepancy may be explained by a mild reduction in other L1 subfamilies that is not detectable in whole-testis RNA sequencing due to masking by background RNA. For example, while we could detect L1 upregulation in *Dnmt3C*$^{KO/KO}$ by IF and RNA-seq in sorted spermatogonia (Fig. 1A,C), we could not detect it when performing RNA-seq in P5 whole testis from the same genotypes (Fig. 5B). These findings raise the possibility that although L1MdA expression increases in dKO testes, overall L1 protein levels remain stable due to the downregulation of other L1 subfamilies, which we could not detect. Future efforts using another *Cre* driver line to delete *Nrf1* in meiosis may be able to test the contribution of NRF1 to the observed boost of LINE1 in meiosis. While our findings indicate that a portion of the differential expression pattern between LINE1 and IAPs is likely attributed to activation by NRF1, the question of what additional factors, other than DNA methylation, cooperate with NRF1 remains elusive. Our chromatin profiling data shows considerable loss of repressive marks at TEs at the onset of meiosis correlating with increased NRF1 binding, suggesting that repressive marks, such as H3K27me3, might prevent NRF1 binding to TEs to some extent (Figs. 5I and EV5H). Accordingly, we found that ERVK expression was repressed by the presence of H3K27me3 in likely bivalent and H3K27me3-marked copies, whereas other TE families seemed less affected by H3K27me3 repression (Appendix Fig. S4B). It is possible that the absence of H3K27me3 at unmethylated IAPs prior to meiosis allows NRF1 binding and their reactivation in spermatogonia (Figs. 5I and EV5H). Supporting this idea, NRF1

does not appear to bind TEs in mESCs in the absence of DNA methylation (Domcke et al, 2015), which remain repressed due to an epigenetic switch towards repressive histone marks such as H3K27me3 and H3K9me3 (Walter et al, 2016). This suggests that repressive histone modifications in mESCs may also prevent the reactivation of hypomethylated TEs by preventing the binding of DNA methylation-sensitive TFs. It is also possible that NRF1 requires other pioneer factors to open up the chromatin, and that its contribution to transcription is determined by the combination of the DNA methylation status of its motif and other features of the chromatin at the TE promoter. Moreover, NRF1 activity can also be modulated by posttranslational modifications, such as phosphorylation (Palmer et al, 2019), adding another layer of regulation to its role in TE expression.

In mice, IAPs are the most mutagenic retrotransposon (Gagnier et al, 2019). In addition, IAPs are prone to behave as metastable epialleles, presenting variegated DNA methylation patterns largely defined by the genetic variation affecting trans-acting repressors such as the KRAB- zinc-finger proteins (Kazachenka et al, 2018; Bertozzi et al, 2021). However, all variegated IAP alleles uncovered to date were found to be fully methylated in sperm, highlighting the effectiveness of DNMT3C and its de novo methylation activity targeting TEs in prospermatogonia. Recently, long-read sequencing uncovered LINE1 copy-specific patterns of DNA hypomethylation that correlate with activity and retrotransposition, a finding that could not be made using the "gold standard" whole genome bisulfite sequencing (Gerdes et al, 2023). Perhaps future work in germ cells using these emerging technologies will be able to identify patterns of hypomethylation in single TE copies more prone to escape the genome defense pathways in place and link these patterns with the presence of germline TF binding motifs and TE reactivation. Finally, our data on NRF1 and chromatin modifications adds to a substantial body of work suggesting that evading DNA methylation and hijacking DNA methylation-sensitive TFs is essential to allow TE activity in germ cells. Novel technologies in long-read epigenomics, combined with the identification of more examples of TFs binding TEs in germ cells, may elucidate how TEs escape germline silencing mechanisms and reactivate, with consequences for genome evolution and stability.

# Methods

### Reagents and tools table

| Reagent/resource | Reference or source | Identifier or catalog number |
|---|---|---|
| **Experimental models** | | |
| *Dnmt3C<sup>KO/KO</sup>* (*M. musculus*) | Barau et al, 2016 | |
| *Oct4-eGFP* (*M. musculus*) | Jackson Laboratories | stock 008214 |
| C57BL/6 (*M. musculus*) | Jackson Laboratories | |
| *Nrf1<sup>fl/fl</sup>* (*M. musculus*) | Wang et al, 2017 | |
| *Ddx4<sup>Cre</sup>/Vasa<sup>Cre</sup>* (*M. musculus*) | Wang et al, 2017 | |

| Reagent/resource | Reference or source | Identifier or catalog number |
|---|---|---|
| *Dnmt3C<sup>KO/KO</sup>; Nrf1<sup>cKO/KO</sup>* (*M. musculus*) | This study | |
| **Antibodies** | | |
| IgG control (1:50) | Abcam | ab37415 |
| Anti-Rabbit IgG (1:100) | Antibodies | ABIN101961 |
| Anti-Mouse IgG (1:100) | Antibodies | ABIN101785 |
| H3K27me3 (C36B11) (1:50) | Cell Signalling | 9733S |
| H3K27ac (1:50) | Abcam | ab4729 |
| H3K9me2 (1:50) | Abcam | ab4729 |
| H3K4me3 (1:50) | Merck Millipore | 07-473 |
| H3K9me3 (1:50) | Abcam | ab8898 |
| NRF1 (1:50) | Abcam | ab34682 |
| NRF1 (1:600 immunofluorescence) | Abcam | ab175932 |
| L1-ORF1p (1:400 immunofluorescence, 1:1000 Western blot) | Abcam | ab216324 |
| TRA98 (1:500) | Abcam | ab82527 |
| POL II (phospho S5) (1:1000) | Abcam | ab5131 |
| α-TUBULIN (1:10000) | Sigma | T5168 |
| IAP-POL (1:400 immunofluorescence, 1:1000 Western blot) | produced by Boulard lab (see "Methods") | Not commercially available |
| CD326 (Ep-CAM)-Alexa Fluor647 (10 µg per 10 mio cells) | Biolegend | 118212 |
| β-2-Microglobulin-PE (10 µg per 10 mio cells) | Santa Cruz Biotechnology | SC-32241 |
| Alexa Fluor 647 (1:800) | ThermoFischer | N/A |
| Alexa Fluor 488 (1:800) | ThermoFischer | A-21208 |
| Hoechst 3342 (2 µg per 10 mio cells) | ThermoFischer | 62249 |
| Propidium Iodide (1 µg per 10 mio cells) | Invitrogen | P3566 |
| **Oligonucleotides and other sequence-based reagents** | | |
| Cloning | This study | Table EV1 |
| Genotyping PCR primers | This study | Table EV1 |
| **Chemicals, enzymes, and other reagents** | | |
| Bovine Serum Albumin (BSA) | Sigma-Aldrich | A0336 |
| Collagenase A | Sigma-Aldrich | 11088793001 |
| Concanavalin A-coated beads | Polysciences | 86057-10 |

| Reagent/resource | Reference or source | Identifier or catalog number |
|---|---|---|
| cOmplete, Mini, EDTA-free Protease Inhibitor Cocktail (PIC) | Roche | 11836170001 |
| DirectZol RNA MicroPrep kit | Zymo | R2061 |
| Digitonin | Millipore / Merck | 300410 |
| DNA Clean & Concentrator kit | Zymo | D4014 |
| Fetal Bovine Serum (FBS) | PAN-Biotech GmbH | N/A |
| Food (for mice) | ssniff | N/A |
| HiTrap Q HP column | Cytiva | 17115401 |
| MssI CpG methyltransferase | NEB | M0226M |
| NHS-activated agarose beads | Pierce | 26196 |
| Nitrocellulose membranes | ThermoFischer | 88018 |
| pMiniT 2.0 Vectors | NEB | E1203S |
| ProLong Gold | Life Technologies | P36930 |
| protein A-Tn5 transposase | Diagenode | C01070001 |
| Protino Ni-NTA column | Macherey-Nagel | 745415.1 |
| SDS-PAGE | GeneScript | M00653, M00654 |
| Streptavidin magnetic beads | Invitrogen Dynabeads MyOne | 650032 |
| SuperSignal West Pico, Femto | ThermoFischer | 1863096 |
| Tryple Epress Enzyme | Fisher Scientific | 12-605-010 |
| **Software** | | |
| bamCoverage v.2.27.1 | https://deeptools.readthedocs.io/en/latest/ | |
| Bedtools v2.29.2 | https://bedtools.readthedocs.io/en/latest/ | |
| Bowtie2 v2.5.3 | https://bowtie-bio.sourceforge.net/bowtie2/index.shtml | |
| ChIPpeakAnno v3.34.1 | https://bioconductor.org/packages/release/bioc/html/ChIPpeakAnno.html | |
| ChIPseeker v1.38.0 | https://bioconductor.org/packages/release/bioc/html/ChIPseeker.html | |
| deepTools v.3.1.0 | https://deeptools.readthedocs.io/en/latest/ | |
| DESeq2 v.1.44.0 | https://bioconductor.org/packages/release/bioc/html/DESeq2.html | |
| edgeR | | |
| FastQC v.0.12.1 | https://www.bioinformatics.babraham.ac.uk/projects/fastqc/ | |
| FlowJo | BD Biosciences | |

| Reagent/resource | Reference or source | Identifier or catalog number |
|---|---|---|
| Graphpad Prism | Graphpad Prism | |
| ImageJ 1.53o version | | |
| MACS2 v2.1.2 | https://github.com/macs3-project/MACS | |
| MaxQuant v.1.6.5.0 | | |
| R v4.2.2, v.4.4.3 | https://www.r-project.org/ | |
| RepEnrich2 | | |
| STAR | https://github.com/alexdobin/STAR | |
| TEtranscripts v2.2.3 | https://github.com/mhammell-laboratory/TEtranscripts | |
| TElocal v1.1.1 | https://github.com/mhammell-laboratory/TElocal | |
| TrimGalore v.0.6.10 | | |
| **Other** | | |
| 2100 Bioanalyzer | Agilent Technologies | |
| Illumina NextSeq 2000 | Illumina | |
| Invitrogen Bigfoot cell sorter | Invitrogen | |
| Stellaris 8 FALCON confocal microscope | Leica Microsystems | |
| Qubit 4.0 Fluorometer | Invitrogen | |

## Mice

Animal experiments were performed according to local and European animal welfare laws and approved by authorities (ethical committees on animal care and use of the federal states of Rheinland-Pfalz, Germany. Covered by LUA license G 23-5-049). Mice (7–24 weeks old) were housed with enrichment material in ventilated cages (humidity 40–70%, temperature $22 \pm 2\,°C$) on a 12 h light/dark cycle and fed ad libitum. For tissue and embryo collection, mice were euthanized by cervical dislocation. Prenatal testis collections were obtained from embryos by following timed pregnancies, where the first day postcoitum was considered E0.5. Postnatal time points were measured starting at birth, which was considered day P1. The mouse strains used in this study were all previously described and bred onto a C57Bl6/J background ($Dnmt3C^{KO/KO}$, $Nrf1^{fl/fl}$, $Ddx4^{Cre}/Vasa^{Cre}$, and $Oct4$-eGFP (Jackson Laboratories, stock 008214)). To generate a $Nrf1$ germ cell conditional Knockout in spermatogonia, $Nrf1^{fl/fl}$ animals were crossed with $Vasa^{Cre}$ animals, respectively. To this end, $Nrf1^{fl/KO}$ female animals were crossed with $Nrf1^{fl/WT;}$ $Ddx4^{Cre}$ male animals. $Nrf1^{fl/KO}$; $Ddx4^{Cre}$; $Dnmt3C^{KO/KO}$ mice were obtained by crossing $Nrf1^{fl/KO}$; $Dnmt3C^{KO/KO}$ female animals with $Nrf1^{fl/WT}$; $Vasa^{Cre}$; $Dnmt3C^{KO/WT}$ males (Appendix Fig. S8A). All offspring were genotyped by PCR (Appendix Fig. S8C) using the primers listed in Table EV1. No blinding was performed in this study on mouse genotypes.

## Immunofluorescent analysis

For cryosections, testes at E15.5, E18.5, P5, P10, P15, P20 were dissected. Embryonic testes were fixed for six h and postnatal testes

overnight in 4% paraformaldehyde at 4 °C, followed by two washes in 1× PBS for 5 min. Two sequential overnight incubations at 4 °C were performed in 15% and 30% sucrose, respectively. Testes were embedded in O.C.T. compound (FSC 22 Frozen Section Compound, Leica), 10-μm sections were cut and spotted onto Superfrost Plus slides (Epredia). For immunofluorescence, slides were calibrated at room temperature (RT), marked with a hydrophobic pen (ImmEdge Pen, VectorLabs), washed in 1× PBS for 10 min, and the blocked and permeabilized (10% donkey serum (Sigma), 3% BSA, 0.3% TritonX-100, 1× PIC in PBS) for 1h at RT. Sections were incubated with primary antibodies (Reagents and Tools table) overnight at 4 °C in blocking buffer, followed by three 1× PBST (0.1% Tween-20 in 1× PBS) washes The sections were incubated for 1h at RT with donkey anti-rabbit Alexa Fluor 647, donkey anti-rat Alexa Fluor 488 secondary antibodies (ThermoFischer) at 1:800 dilution and DAPI (2 μg/μl), to stain nuclei. Three 1× PBST washes followed, and the sections were mounted with ProLong Gold antifade mountant (Life Technologies). All images were acquired with a Stellaris 8 FALCON confocal microscope (Leica Microsystems, Wetzlar, Germany) equipped with a White light Laser (WLL). In more detail, images (bidirectional mode, 400 Hz scan speed) were acquired sequentially using 63×/1.4NA oil immersion objective and HyD detectors in photon counting mode. Excitation lines, 405, 488, and 640 nm, and emission bands 430–470, 500–600, and 650–750 nm for DAPI, Alexa Fluor 488-TRA98 and Alexa Fluor 647-L1-ORF1, respectively. The pixel size was set to 48 nm for the images that were used for quantification. All images were processed with Fiji ImageJ (1.53o version) (Schneider et al, 2012).

## Quantification analysis in ImageJ

Quantification of TRA98 and L1-ORF1 signals in the nucleus and cytoplasm was performed using images (256 × 256 pixels) of selected cells that were manually cropped from a larger field of view. Nuclei were segmentated using the DAPI channel (Fig. EV1C). First, the Gaussian filter (sigma value 6) was applied, and a threshold was set based on OTSU algorithm (Otsu, 1979). The segmentation was fine-tuned by filling holes and processing a watershed operation to split touching nuclei and create the region of interest (ROI) for the nucleus. For the cytoplasm ROI, a ring ROI around the nucleus was created by dilating the nucleus ROI ten times and subtracting out the nucleus ROI. Mean intensity in TRA98 and L1-ORF1 channels were measured for both the nucleus and ring ROI. The number of nuclei in whole sections was calculated using the Analyze Particles tool after applying a threshold in ImageJ (Fig. EV5A,B). The bar blot was generated using GraphPad Prism v10 (Fig. EV5B). For Appendix Fig. 8, quantification of TRA98 and L1-ORF1 signals in the nucleus and cytoplasm was performed using (256 × 256 pixels, 4 z-planes) of selected cells that were manually cropped from a larger field of view. From the z-stack of four planes, the slice with the highest mean intensity in DAPI channel was used for analysis. Nuclei was segmented using the DAPI channel (Fig. 5E) using local threshold (method Sauvola, radius 50, parameter 1 = 0.25, parameter 2 = 0.5) (Sauvola and Pietikäinen, 2000). The segmentation was fine-tuned by filling holes, processing watershed operation to split touching nuclei, filtering out objects smaller than 10 μm² and create the region of interest (ROI) for the nucleus. For the cytoplasm ROI, a ring ROI around the nucleus was created by dilating the nucleus

ROI 10 times and subtracting out the nucleus ROI. Mean intensity in TRA98 and L1-ORF1 channels was measured for both the nucleus and ring ROI.

## Western blot

For western blots, protein samples were extracted by homogenizing testis in RIPA buffer supplemented with cOmplete, Mini, EDTA-free Protease-inhibitor-cocktail (PIC) (Roche, 11836170001). The samples were sonicated for 5 min (30 s ON, 30 s OFF) and centrifuged for 30 min at 16,000 × g at 4 °C. The total protein concentration was quantified by the Bradford assay. In total, 15 μg of protein per sample were separated on SDS-PAGE (GeneScript, M00653, M00654) and transferred to nitrocellulose membranes (ThermoFischer, 88018). Membranes were blocked for 40 min with 5% BSA in TBST and incubated with primary antibodies (Reagents and Tools table) overnight at 4 °C. The membranes were washed with TBST and incubated with HRP-conjugated secondary antibodies for 1 h at RT. The protein bands were visualized using chemiluminescence reagents (SuperSignal West Pico, Femto, ThermoFischer, 1863096).

## Cell sorting by flow cytometry

Prospermatogonia were isolated from E15.5 Oct4-eGFP male mice testes and collected in 1× PBS. Testes were decapsulated and transferred to tubes containing 100 μl collagenase solution in HBSS (Collagenase A, 2× AAs, 2× Na-pyruvate, and 25 mM HEPES-KOH, pH 7.5) and dissociated at 37 °C for 8 min. After adding 200 μl TrypLE Express, the cell suspension was incubated for 5 min at 37 °C and pipetted to obtain a single-cell suspension. TrypLE was quenched with 70 μl pre-warmed fetal bovine serum (FBS), and cells were washed twice with FACS buffer (PBS, 2 mM EDTA, 25 mM HEPES-KOH, 1.5% BSA, 10% FBS) after centrifugation at 600 × g for 4 min. Cells were resuspended in DAPI-supplemented (2 μg/mL) FACS buffer, filtered through 40-μm cell strainers, and sorted using an Invitrogen Bigfoot with a 100-μm nozzle. DAPI-negative and OCT4-eGFP-positive cells were sorted into low-binding tubes containing 100 μl PBS (Appendix Fig. S2A).

Spg were isolated from P14 mice testes and processed similarly. Single-cell suspensions were stained with 8.25 mg per 5 million cells of CD326 (Ep-CAM)-Alexa Fluor® 647 (BioLegened, 118212) and ß-2-Microglobulin-PE (Santa Cruz Biotechnology, SC-32241) for 20 min on ice, washed with FACS Buffer twice, and resuspended in DAPI-supplemented (1 μg/mL) FACS Buffer, then sorted using the same conditions (Appendix Fig. S1A,B). DAPI-negative, ß-2-Microglobulin-negative, and Ep-CAM-positive cells were collected in 100 μl PBS.

Spc were isolated from the testes of mice older than 20 days. Testes were decapsulated and dissociated in 2 mL collagenase solution (40 U of Collagenase A, 1.5× AAs, 1.5× Na-pyruvate, and 25 mM HEPES-KOH, pH 7.5) at 37 °C. The cell suspension was sedimented for 2 min, and the supernatant was aspirated. After treating with 250 μl TrypLE and filtering, 10 million cells were stained with 0.5 μL Hoechst 3342 (Thermo Scientific, 62249) for 8 min. TrypLE was quenched by FBS followed by a centrifugation, cells were resuspended in buffer supplemented with 1 μL Hoechst 3342 per 10 million cells at 35 °C for another 8 min, filtered, and stained with Propidium Iodide (Invitrogen, P3566). Cells were sorted using the same conditions by Propidium Iodide exclusion and Hoechst 3342 staining, indicating 2N DNA content (Appendix Fig. S1C,D).

## CUT&Tag

Experiments were conducted on freshly FACS-isolated germ cells in duplicates. CUT&Tag was performed following the published protocol with minor modifications (Kaya-Okur et al, 2020). Briefly, 10,000–20,000 cells were bound to 10 μL Concanavalin A-coated beads (Polysciences, 86057-10) and incubated with primary antibodies specific to the protein of interest, listed in the Reagents and Tools table. Following binding with a secondary antibody, protein A-Tn5 transposase (Diagenode, C01070001) was added 1:400 to introduce sequencing adapters at target sites. The Wash Digest Buffer was supplemented with 0.05% Digitonin for prospermatogonia and spermatogonia and with 0.025% for spermatocytes. DNA was extracted using the DNA Clean & Concentrator kit (Zymo, D4014) and libraries were amplified by PCR using 15 amplification cycles. Pooled libraries were sequenced on an Illumina NextSeq 2000 sequencing device as 2× 155-nt paired-end reads, yielding 7–15 million read pairs per sample.

## CUT&Tag analysis and visualization

Reads were trimmed with TrimGalore (v.0.6.10) to remove Illumina adapter sequences with a minimum read length of 10 bp, and quality reports were generated using FastQC (v.0.12.1). Trimmed reads were mapped to the Mus musculus genome mm10 using Bowtie2 (v.2.5.3) (Langmead and Salzberg, 2012) in local alignment mode allowing a minimum fragment length of 10 bp and a maximum of 1000 bp, while excluding mixed and discordant alignments. Multimapping reads were assigned randomly as previously described (Teissandier et al, 2019). For chromatin modifications in spermatogonia and spermatocytes, uniquely mapped reads were selected based on a mapping quality score of 2:10 using Samtools (version 1.10). Since CUT&Tag fragments can share identical starting and ending positions because the Tn5 integration sites are affected by DNA accessibility, duplicates were retained. The bigWig files were generated from the BAM files using bamCoverage (v.2.27.1), with normalization set to RPGC. After initial quality control and comparing metaplots on all genes, replicates from CUT&Tag experiments were merged as BAM files. Samples were normalized to genome coverage using RPGC. DNMT3C targets were defined as differentially methylated regions from published WGBS data (Barau et al, 2016). These regions were overlapped with the RepeatMasker annotation (open-4.0.5-Repeat Library 20140131), selecting only those with LINEs (L1) longer than 5000 bps and LTRs longer than 500 bps using ChIPseeker (v1.38.0). Downstream visualization of reads as heatmaps or metaplots were generated using deepTools (v.3.1.0).

Peak calling was performed using MACS2 (v.2.1.2) on BAM files (parameters: -g mm --broad), specifying the corresponding IgG sample as control for chromatin marks (Fig. 2B) and no control for TFs (Figs. 4D and EV4E). To identify likely bivalent, H3K4me3-marked, H3K27me3-marked and Uncategorized clusters of DNMT3C targets, called peaks from H3K4me3 and H3K27me3 CUT&Tag in $Dnmt3C^{KO/KO}$ were overlapped with DNMT3C targets using bedtools (v.2.29.2). Peaks commonly called in H3K4me3 and H3K27me3 datasets overlapping with DNMT3C targets were called likely bivalent, peaks overlapping with only one of the two datasets and DNMT3C targets were called H3K4me3-marked or H3K27me3-marked, respectively. DNMT3C targets not overlapping with any peaks from both datasets were called Uncategorized. Venn diagrams and peak overlaps from NRF1 CUT&Tag datasets with DNMT3C targets were analyzed and visualized using ChIPpeakAnno (v.3.34.1) using a maxgap of 750 bp.

RepEnrich2 was run separately on NRF1 and IgG control datasets obtained from $Dnmt3C^{KO/KO}$ and wild-type Spg and Spc. This allowed for the quantification of raw read counts mapping to retrotransposons. To identify differentially enriched transposons, we utilized edgeR, comparing NRF1 enrichment in $Dnmt3C^{KO/KO}$ vs. wild-type across stages and NRF1 binding vs. IgG control in $Dnmt3C^{KO/KO}$ and wild-type Spg and Spc. Normalization and differential analysis were performed using edgeR's trimmed mean of M-values (TMM) method. Log$_2$ fold-change values and statistical significance were computed using a negative binomial model with Benjamini-Hochberg correction for multiple testing. Data visualization was conducted in R (v.4.4.3) using plotMD to generate MA plots visualizing global and significant changes in TE enrichment and the ggplot2 package to generate bar plots on log$_2$ fold-change enrichment.

## RNA extraction and library preparation

Sorted germ cells or testes of littermates were collected and frozen in Trizol until triplicates of each genotype were collected. RNA extraction was performed using the DirectZol RNA MicroPrep kit (Zymo, R2061).

NGS library prep was performed with Illumina's Stranded mRNA Prep Ligation Kit following Stranded mRNA Prep Ligation ReferenceGuide (April 2021) (Document 1000000124518 v02). Libraries were prepared with a starting amount of 10 ng for RNA extracted from sorted Spg and Spc and 29 ng for RNA extracted from whole testes mutants and amplified in 13 PCR cycles and 12 PCR cycles, respectively. Two post PCR purification steps were performed to exclude residual primer and adapter dimers. Libraries were profiled in a DNA High Sensitivity chip on a 2100 Bioanalyzer (Agilent technologies) and quantified using the Qubit 1× dsDNA HS Assay Kit, in a Qubit 4.0 Fluorometer (Invitrogen by Thermo Fisher Scientific).

16 Samples from sorted Spg and Spc and 17 samples from whole testes mutants were pooled in equimolar ratio aiming for at least 50M reads per sample, and sequenced on an Illumina NextSeq 2000 sequencing device with 1 dark cycle upfront R1 and R2 as 2× 108-nt paired-end reads, yielding 50–60 million read pairs per sample.

## RNA-sequencing analysis

One replicate from all genotypes was excluded for RNA sequencing on spermatogonia and on whole testes P5 due to poor RNA quality, as indicated by RIN values below 8. Reads were aligned on the Mus musculus genome assembly GRCm38 (GENCODE release M25) using STAR (v.2.7.3a, parameters: --outMultimapperOrder Random --outSAMattributes NH HI AS nM MD –outSJfilterReads All –outSAMunmapped Within –outFilterMismatchNoverReadLmax 0.04 –outFilterMismatchNmax 999 --winAnchorMultimapNmax 100 --outFilterMultimapNmax 100) (Dobin et al, 2013). The TEtranscripts program (v.2.2.3, parameters: --mode multi --sortByPos --stranded reverse) was used to annotate reads to both genes and

transposable elements (Jin et al, 2015). Differentially expressed analysis was determined using DESeq2 (v.1.44.0) with corrected *P* value (Benjamini and Hochberg, FDR) <0.01 (Love et al, 2014). TElocal (v.1.1.1, parameters: --mode multi --sortByPos --stranded reverse) was used to quantify transposable element expression at the locus level, and the counts were normalized to TPM (Transcripts Per Million) values. For each developmental stage, the z-score was computed by comparing the average log(TPM) of that stage to the mean and standard deviation of log(TPM) across all stages.

## In vitro DNA methylation and bisulfite cloning

The sequences of the transposon promoters from IAP LTR1a and the first two monomers from L1MdT were synthesized (IDT DNA gBlocks), cloned, and PCR-amplified using the primers listed in Table EV1. To generate sequence-unspecific scrambled controls, the CpG sites of the transposon promoters were preserved, but the rest of the sequences were shuffled, then synthesized, amplified, and biotinylated similarly (Fig. EV3A,B). Biotinylated probes were in vitro methylated using the MssI CpG methyltransferase kit (NEB, M0226M) with a ratio of 0.8 µL enzyme per 2 µg DNA. Methylation levels were checked by bisulfite conversion (EZ DNA Methylation Quick kit, D5005), amplification, cloning into pMiniT™ 2.0 Vectors (NEB, E1203S), followed by transformation and Sanger sequencing of ten clones per probe (Fig. EV3C,D).

## DNA pulldown and mass spectrometry of proteins

Nuclear proteins were extracted from pooled testes of 20 P15 male mice in triplicate. Testes were homogenized in Cytoplasmic Buffer (10 mM HEPES-KOH pH 7.5, 10 mM KCl, 1.5 mM MgCl$_2$, 1 mM DTT, 1× PIC) and incubated on ice for 30 min. After centrifugation at 1100 × $g$ for 10 min at 4 °C, the nuclear pellet was resuspended in Nuclear Soluble Buffer (20 mM HEPES-KOH pH 7.9, 420 mM NaCl, 2 mM MgCl$_2$, 20% glycerol, 1 mM DTT, 0.2% IGEPAL CA-630, 1× PIC), vortexed, and incubated for 30 min at 4 °C. The insoluble fraction was pelleted by centrifugation at 14,000 × $g$ for 20 min at 4 °C. The pellet was resuspended in Nuclear Soluble Buffer, sonicated (10 cycles, 10s on/50s off), and centrifuged again. The solubilized fraction was combined with the soluble nuclear extract and the total protein concentration was quantified by Bradford assay.

As described in (Harris et al, 2018), 60 µg of bait DNA was coupled to streptavidin magnetic beads (Invitrogen Dynabeads MyOne, 650032) in coupling buffer (20 mM Tris-HCl pH 8.0, 2M NaCl, 1 mM EDTA, 0.1% Tween-20). Beads were rotated for 1 h at RT and resuspended in Pull Down Buffer (50 mM Tris-HCl pH 8.0, 150 mM NaCl, 5 mM MgCl$_2$, 0.5% IGEPAL CA-630, 1 mM DTT, 1× PIC), then incubated with 250 µg protein extracts for 3 h at 4 °C. After three washes, proteins were eluted in LDS buffer with DTT, boiled at 80 °C for 10 min, and submitted for mass spectrometry (MS) preparation.

## MS sample preparation and measurement

Protein samples were separated by gel electrophoresis on a 4–12% NuPAGE Novex Bis-Tris precast gel (Thermo) at 180 V in 1× NuPAGE MES buffer (Thermo Scientific) for 8 min. After protein fixation (7% acetic acid, 40% methanol) and Coomassie G250 blue staining the background was destained using water. Individual lanes were cut, minced and destained using 50% EtOH/25 mM ammonium bicarbonate buffer pH 8.0 (ABC). The gel pieces were dehydrated with 100% acetonitrile (ACN) and afterwards incubated with reduction buffer (10 mM DTT in 50 mM ABC) at 56 °C for 60 min. Alkylation was performed with iodoacetamide (50 mM IAA in 50 mM ABC) for 45 min at RT in the dark. Gel pieces were completely dehydrated with ACN, dried, covered with trypsin solution containing 1 µg MS-grade trypsin (Sigma) per sample and incubated overnight at 37 °C. Subsequently, peptides were extracted by incubation with the extraction buffer (30% ACN) and dehydration with 100% ACN. ACN was completely removed in a concentrator (Eppendorf). Peptides were de-salted on StageTips (Rappsilber et al, 2007). Tryptic peptides were separated on a nanocapillary (New Objective) packed with Reprosil C18 (Dr. Maisch) using an Easy nLC 1000 system (Thermo). Peptides were eluted from the column with a 90 min gradient from 2 to 60% ACN with 0.1% formic acid at a flow rate of 225 nL/min. The HPLC system was directly connected to a Q Exactive Plus mass spectrometer (Thermo). The mass spectrometer performed HCD fragmentation with a data-dependent Top10 MS/MS spectra acquisition scheme per MS full scan in the Orbitrap analyzer. Data was analyzed with MaxQuant (1.6.5.0) [(Cox and Mann, 2008)] using a uniport mouse database (54,220 entries) and standard settings, except match between runs and LFQ quantitation were activated. Contaminants, reverse hits and proteins identified by site only were removed, and protein groups were filtered for two peptides (one of them unique). Further data processing and graphical representations were done with the R framework.

## Motif analysis using AME

High confidence transposon annotations that did not overlap with annotated genes were generated using the pipeline from (Sakashita et al, 2020). For our analysis, we focused on LINE, IAP and ERV families that were subsetted from this using the grep tool. TEs from these families were intersected with DNMT3C differentially methylated regions to focus on DNMT3C targets (Barau et al, 2016). Motif enrichment was done using the AME suit programme from MEME (McLeay and Bailey, 2010), and transcription factors that had enrichment for DNMT3C targets vs. all transposons were considered. Database used for TF motifs was HOCOMOCOv11 (HOCOMOCOv11_full_MOUSE_mono_meme_format.meme). Hits from this were intersected with proteins enriched in IP-MS using their Uniprot identifiers (Fig. 3A; Dataset EV2 (AME data)).

## Generation of antibodies anti-IAP-POL

The polyclonal antibodies anti-IAP-POL were generated by immunization against a recombinant murine Intracisternal A-particle retrotransposons POL protein produced in *Escherichia coli*. The cDNA encoding IAP-POL (Uniprot P12894) was synthesized (IDT), subcloned into the pETM11-SUMO3 expression vector (EMBL) which was then transformed into *E. coli* BL21-CodonPlus(DE3)-RIL cells. Precultures were grown at 37 °C in LB medium supplemented with 30 µg/mL Kanamycin and 34 µg/mL Chloramphenicol and used to inoculate large-scale expression cultures in TB-FB medium supplemented with 1.5% lactose, 30 µg/mL Kanamycin and 34 µg/mL Chloramphenicol. The cultures were grown overnight at 25 °C under agitation, harvested by centrifugation (30 min, 5000 × $g$, 4 °C), and the pellets were flash-frozen in

liquid nitrogen and stored at $-80\,°C$. The cell pellet was then resuspended in a lysis buffer containing 50 mM Tris-HCl pH 8.0, 250 mM NaCl, 20 mM imidazole, 1 µg/mL DNaseI, 5 mM MgCl$_2$ and EDTA-free PIC (Roche). After the addition of 1 mg/mL lysozyme and incubation at RT for 30 min, the cells were lysed using a microfluidizer, followed by centrifugation (30 min, 140,000 $\times$ $g$, $4\,°C$). The pellet was washed three times in PBS with 0.01% Triton X-100 and resuspended overnight at RT in PBS supplemented with 6 M guanidinium hydrochloride. The mixture was sonicated and centrifuged (20 min, 140,000 $\times$ $g$, $4\,°C$), the supernatant was loaded onto a 5 mL Protino Ni-NTA column (Macherey-Nagel) pre-equilibrated with 50 mM Tris-HCl pH 8.0, 250 mM NaCl, 20 mM imidazole, 6 M ureum and 1 mM DTT. The column was washed with the equilibration buffer (EB) and then eluted with the EB supplemented with 300 mM imidazole. Eluted fractions were analyzed by SDS-PAGE and the fractions containing the His6-SUMO3-IAP-POL fusion protein were pooled and dialyzed overnight at $4\,°C$ against a buffer containing 50 mM Tris-HCl pH 8.0, 250 mM NaCl, 20 mM imidazole, 2 M ureum and 1 mM DTT. To cleave the His$_6$-SUMO3 tag, a His6-tagged SenP2 protease was added to the sample. To remove the His6-SUMO3 tag, the sample was loaded again onto a 5 mL Protino Ni-NTA column pre-equilibrated with 50 mM Tris-HCl pH 8.0, 250 mM NaCl, 20 mM imidazole, 2 M ureum and 1 mM DTT. The column was then washed with the respective EB and eluted with EB supplemented with 300 mM imidazole. The flow through fraction containing the untagged IAP-POL protein was dialyzed overnight at $4\,°C$ against the IEX buffer (50 mM Tris-HCl pH 7.0, 50 mM NaCl, 2 M ureum and 1 mM DTT). The dialyzed IAP-POL protein was loaded onto a 5 mL HiTrap Q HP column (Cytiva) pre-equilibrated with IEX buffer, washed with IEX buffer and eluted with a gradient ranging from 50 mM NaCl to 1 M NaCl over 12 column volumes. Purified IAP-POL protein was concentrated to ~1.1 mg/mL, flash-frozen, and stored at $-80\,°C$. The identity and the integrity of IAP-POL were confirmed by mass spectrometry at the EMBL Proteomics Core Facility. Polyclonal antibodies against the purified IAP-POL protein were raised in a New Zealand White rabbit at the EMBL Laboratory Animal Resources facility. Following immunization, the rabbit was sacrificed, and serum was collected from the final bleed for antibody purification. The purified IAP-POL protein was covalently coupled to NHS-activated agarose beads (Pierce) according to the manufacturer's specifications. The serum was diluted 1:1 in PBS and incubated overnight with the IAP-POL resin. The resin was then washed with PBS, and the IAP-POL-specific antibodies were eluted with 100 mM Glycine pH 2.4, 150 mM NaCl and neutralized with 1 M Tris pH 8.5. The elution fractions were analyzed by SDS-PAGE. The fractions containing antibodies were pooled, and the aliquots were mixed with 50% glycerol, flash-frozen and stored at $-70\,°C$.

## Data availability

Additional datasets are summarized in Datasets EV1–3. The sequencing data produced in this publication have been deposited in NCBI's Gene Expression Omnibus (Edgar et al, 2002) and are assigned the identifier GSE282280.

The source data of this paper are collected in the following database record: biostudies:S-SCDT-10_1038-S44319-025-00526-1.

## Peer review information

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

## Acknowledgements

We thank members of JB's laboratory and animal caretakers, S. Schlegel and T Dehn, at IMB. We thank S Möckel and S Nick of the IMB Flow Cytometry Core Facility for their support and assistance with FACS-isolation on the Invitrogen BigFoot Cellsorter (funded by the Deutsche Forschungsgemeinschaft (DFG, German Research Foundation)—511658729). In addition, we thank the IMB Microscopy and Histology Core Facility, especially P Turunen, for their assistance with super-resolution image analysis and image processing. Imaging was performed on the STELLARIS 8 Confocal microscope (funded by the Deutsche Forschungsgemeinschaft (DFG, German Research Foundation) - P#497669232). Support by the IMB Genomics Core Facility and the use of its NextSeq 2000 is gratefully acknowledged. We thank J Scheurich and K Remans, and the EMBL Protein Expression and Purification Core Facility for their critical support with the antibody purification. We gratefully acknowledge advice and technical support by V Morin, C Keller-Valsecchi, MF Basilicata, A Kalita, M Greenberg, SA Prakash, E Parasyraki. JB, JL, and SEK were funded by a German Research Foundation collaborative research centre grant (DFG grant SFB 1361, project number 393547839). AS was supported by the Forschungsinitiative Rheinland-Pfalz (ReALity). MB and MM were supported by the European Molecular Biology Laboratory. YW is partially supported by the US National Science Foundation CAREER Award (IOS 2042908).

## Author contributions

**Jessica Leismann**: Conceptualization; Formal analysis; Investigation; Methodology; Writing—original draft; Project administration; Writing—review and editing. **Styliani-Eirini Kanta**: Formal analysis; Methodology; Writing—original draft; Writing—review and editing. **Ishita Amar**: Formal analysis; Writing—review and editing. **Anna Szczepińska**: Formal analysis. **Monika Mielnicka**: Resources. **Giuseppe Petrosino**: Data curation; Software; Formal analysis; Methodology. **Anke Busch**: Data curation; Software; Formal analysis. **Marion Scheibe**: Resources; Investigation. **Pengxiang Wang**: Resources. **Yuan Wang**: Resources. **Falk Butter**: Resources. **Matthieu Boulard**: Resources; Writing—review and editing. **Joan Barau**: Conceptualization; Resources; Formal analysis; Supervision; Funding acquisition; Methodology; Project administration; Writing—review and editing.

Source data underlying figure panels in this paper may have individual authorship assigned. Where available, figure panel/source data authorship is listed in the following database record: biostudies:S-SCDT-10_1038-S44319-025-00526-1.

## Disclosure and competing interests statement

The authors declare no competing interests.

# Expanded View Figures

**Figure EV1.   Related to Fig. 1.**

(A) Scheme showing percentage of mice testes cells with different age classified into developmental stages of Spg, Spc, spermatids and somatic cells. Meiosis I with Leptotene (L), Zygotene (Z), Pachytene (P) and Diplotene stages as well as Meiosis II are depicted in Spc. This image is modified from (Ernst et al, 2019). (B) Scheme showing the DNA methylation dynamics of wild-type and $Dnmt3C^{KO/KO}$ mouse germ cells across spermatogenesis. (C) Representative immunostaining from biological duplicates of LINE1-ORF1p and TRA98 (germ cell marker) on cryosections of wild-type (WT) and $Dnmt3C^{KO/KO}$ testes at specific developmental time points centered on one representative germ cell nucleus. Zoom10, Scale bar, 5 μm. (D) Box plots displaying the mean intensity of LINE1-ORF1p quantification from three nuclei as in (C). Error bars represent 1.5 times the interquartile range (IQR) above and below the median, where IQR is defined as the difference between the third and first quartiles.

▶

**A**

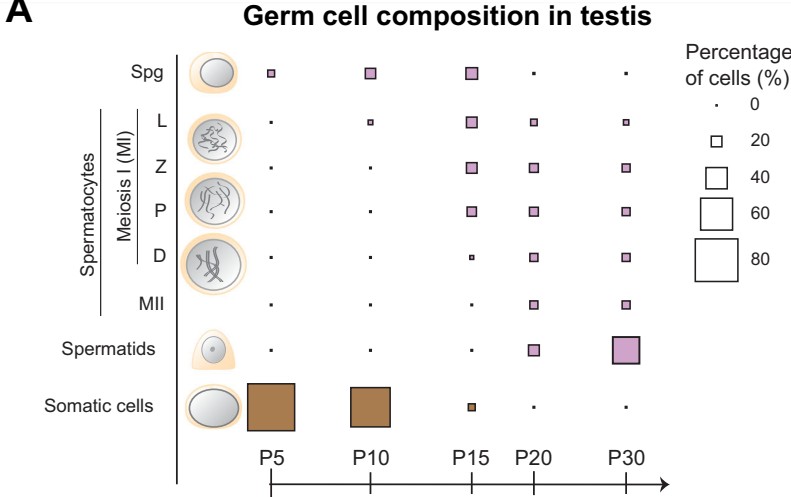

**Germ cell composition in testis**

**B**

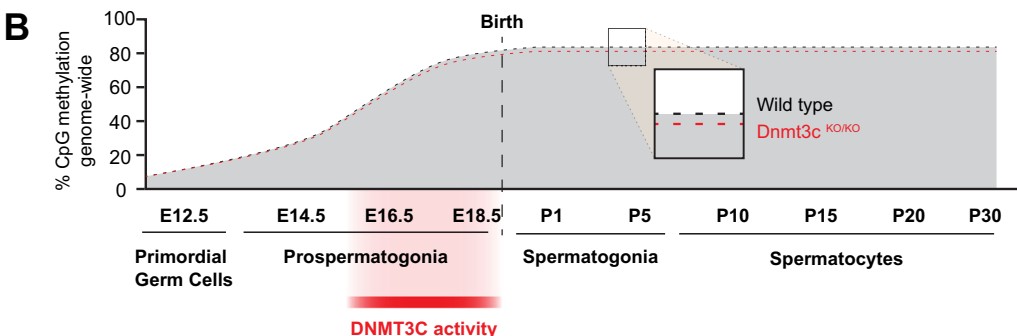

**C**

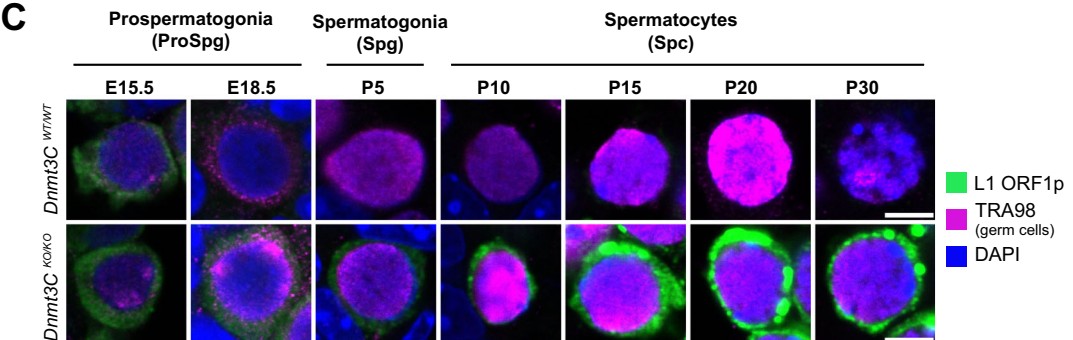

**D**

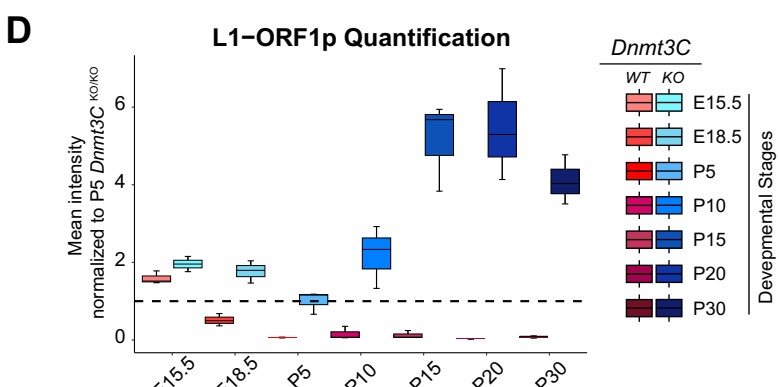

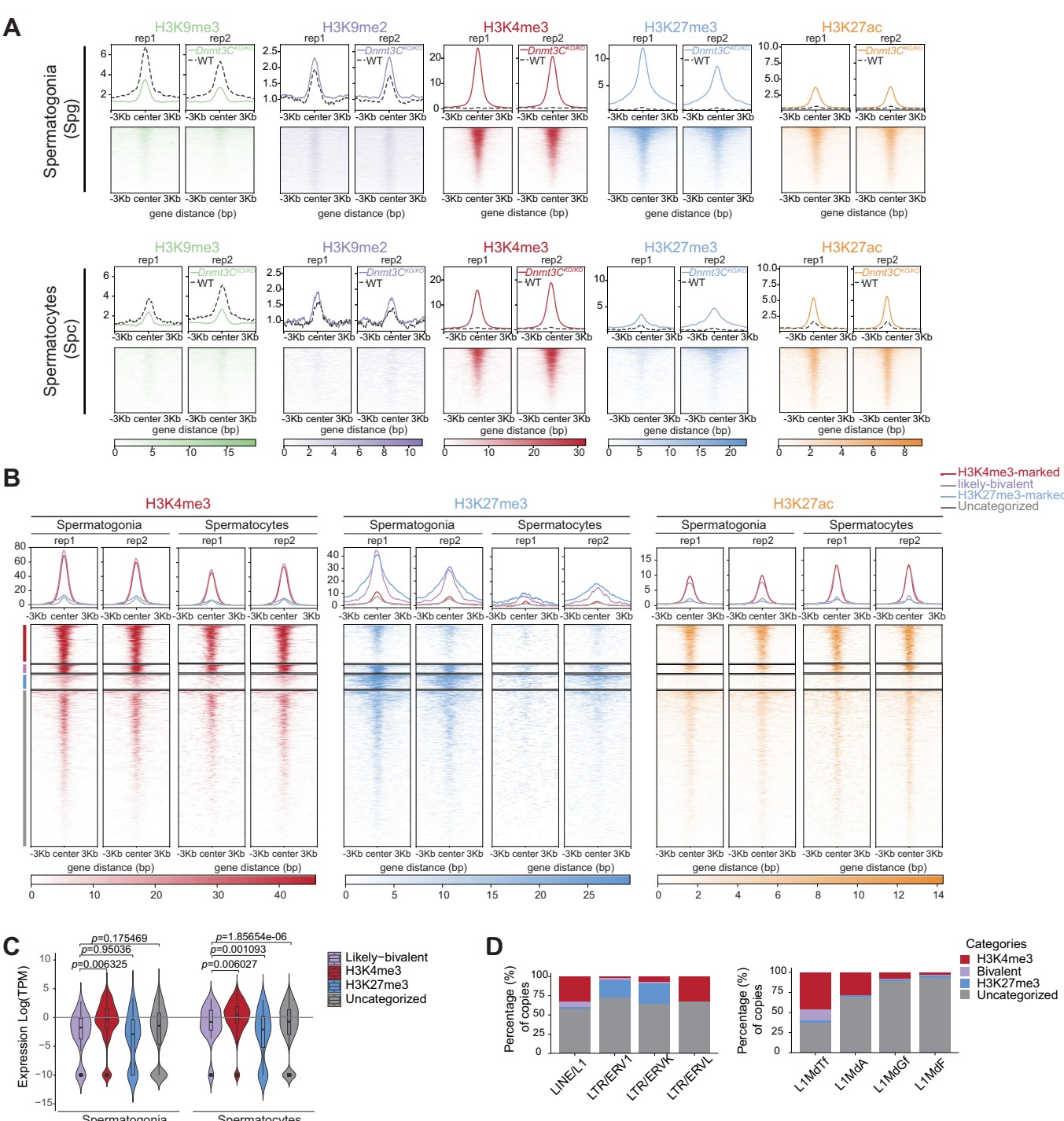

**Figure EV2.  Related to Fig. 2.**

(A) Heatmaps displaying normalized coverage and metaplots showing mean enrichment of H3K4me3, H3K27me3, H3K9me2, H3K9me3 and H3K27ac CUT&Tag from uniquely mapped reads of two biological replicates (rep), centered on DNMT3C targets in sorted *Dnmt3C*^KO/KO Spg and Spc. The corresponding mean enrichment in wild-type is illustrated as a dashed black line in the same metaplot. (B) Heatmaps showing normalized coverage and metaplots displaying mean enrichment of H3K4me3, H3K27me3 and H3K27ac CUT&Tag from uniquely mapped reads centered on likely bivalent, H3K4me3-marked, H3K27me3-marked and uncategorized DNMT3C targets in sorted *Dnmt3C*^KO/KO Spg and Spc. (C) Violin plots with boxplots showing log(TPM) values from RNA sequencing of three biological replicates from TElocal of likely bivalent, H3K4me3-marked, H3K27me3-marked and uncategorized DNMT3C targets from uniquely mapped reads in *Dnmt3C*^KO/KO Spg and Spc. The median and IQR range of the boxplot are defined as in Fig. 2C. P values were tested using an unpaired *t* test between likely bivalent clusters and other clusters of the same stage and genotype. (D) Column charts showing the percentage of TE classes and families classified as likely bivalent, H3K4me3-marked, H3K27me3-marked, uncategorized DNMT3C targets from uniquely mapped reads of two biological replicates.

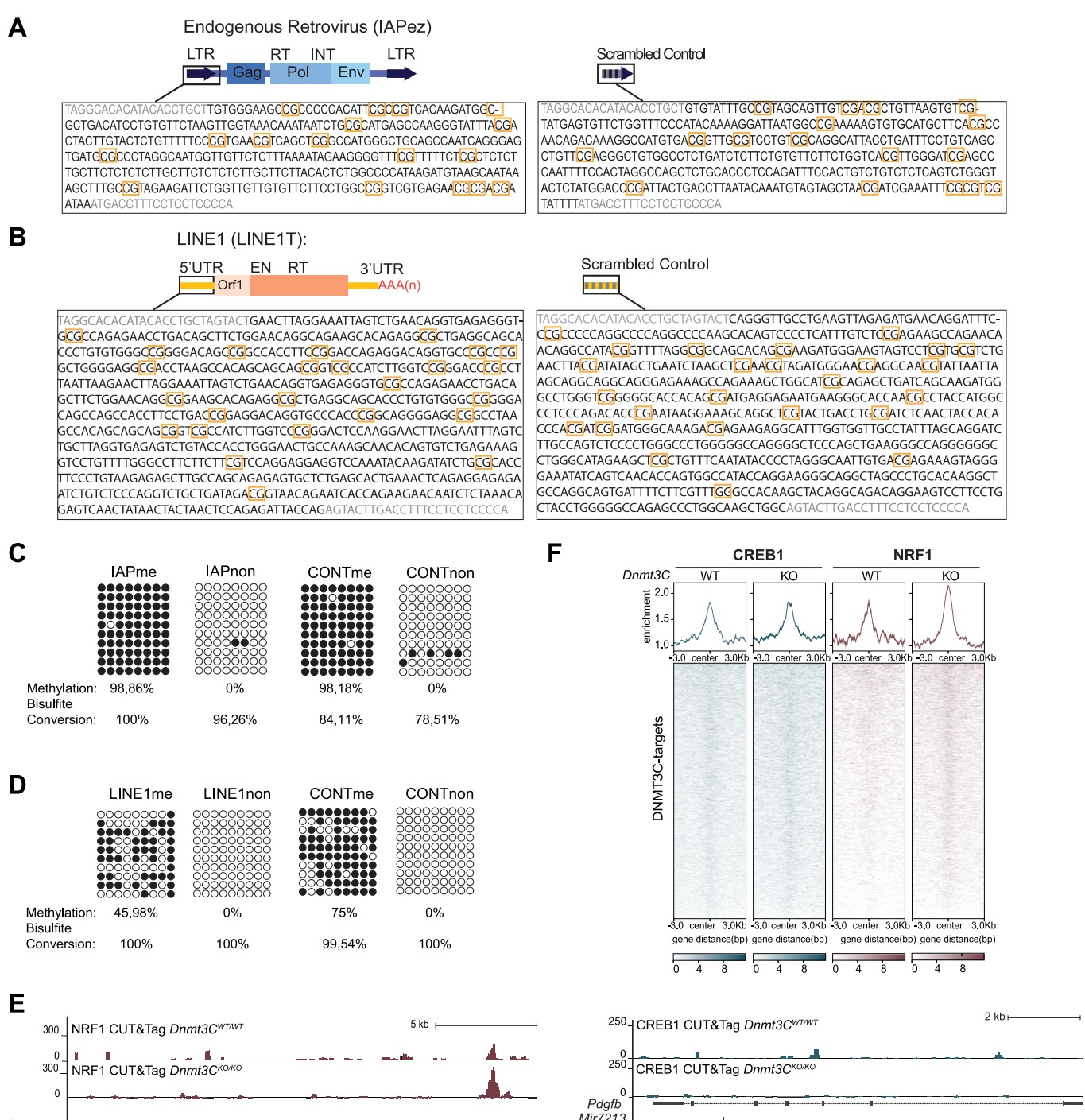

Figure EV3.   Related to Fig. 3.

(A) Illustration of the bait sequence used from the IAP LTR1a promoter, and the corresponding scrambled control used in the DNA-pulldown experiment. CpG sites, which are identical between IAP LTR1a and the scrambled control, are highlighted with a yellow frame. The sequences corresponding to random primers used for amplification and biotinylation, are shown in gray. (B) as in (A) for the LINE1MdT promoter and its scrambled control. (C) Bisulfite cloning results for IAPme, IAPnon, CONTme, and CONTnon, represented as black/white circle diagrams, with black circles indicating methylated cytosines and white l indicating unmethylated. Each column is a CpG site, and each row represents a biological replicate (n=10). Average methylation percentages and bisulfite conversion rates are indicated at the bottom. (D) As in (C) for LINE1me, LINE1non, CONTme and CONTnon. (E) Representative track example of CREB1 and NRF1 CUT&Tag example at bona fide targets *Pdgfb* and *Ccdc155*, respectively. (F) Heatmaps displaying normalized coverage and metaplots showing mean enrichment of CREB1 and NRF1 at DNMT3C targets in wild-type and *Dnmt3C*[KO/KO] Spermatogonia (Spg) merged from two biological replicates.

**A**

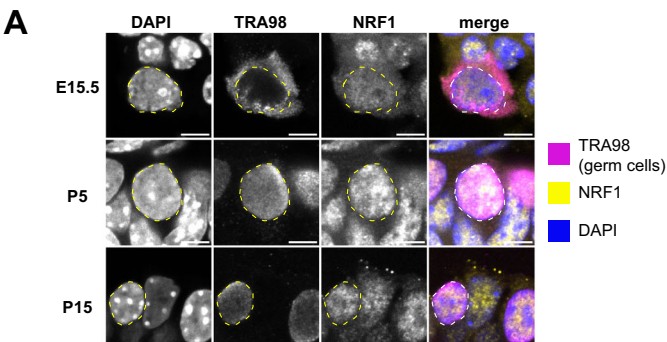

**B**

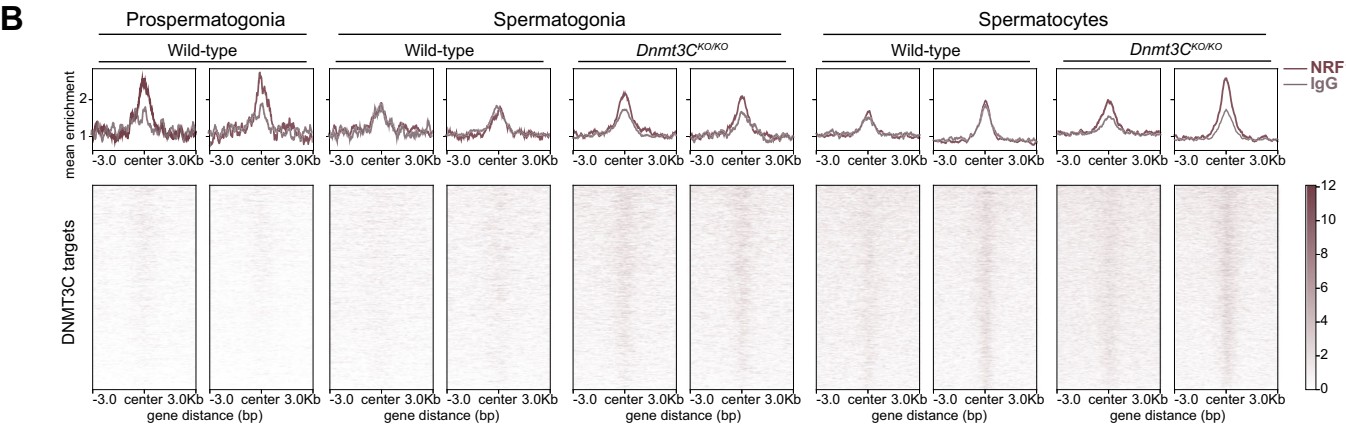

**C**

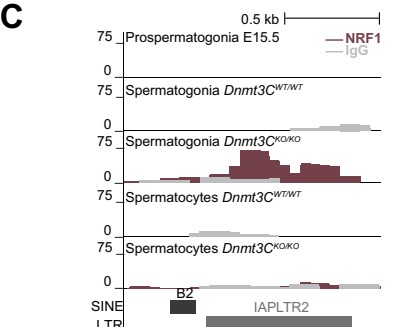

**D**

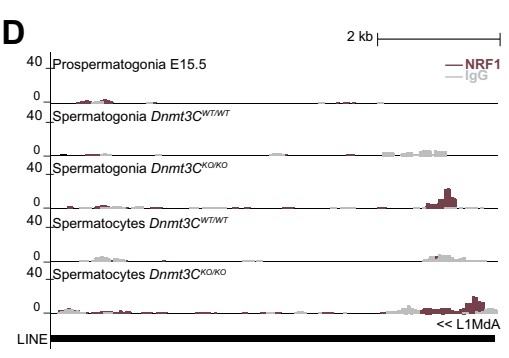

**E**

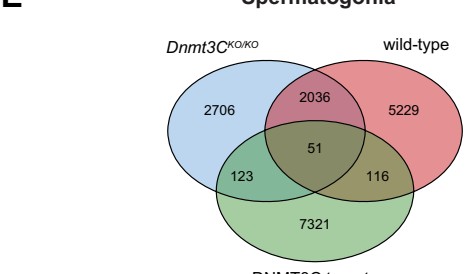

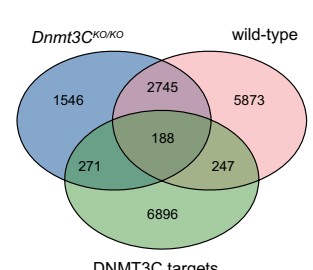

◄ **Figure EV4.  Related to Fig. 4.**

(A) Immunostaining of testes cryosections at E15.5, P5 and P15 showing TRA98 (germ cell marker) and NRF1 in the nucleus from one biological replicate, which is illustrated by DAPI and highlighted with a dashed yellow line. Scale bar 5 μm. (B) Heatmaps displaying normalized coverage and metaplots showing mean enrichment of NRF1 at DNMT3C targets wild-type and *Dnmt3C*$^{KO/KO}$ Spg and Spc from two biological replicates. The corresponding mean enrichment of the IgG control is illustrated as a gray line in the same metaplot. (C) Representative track example of NRF1 CUT&Tag as in (D) at an IAPLTR2 (ERVK) copy located at chr11: 101,391,538–101,402,538. (E) Venn diagrams showing overlaps of NRF1 peaks with DNMT3C targets in wild-type and *Dnmt3C*$^{KO/KO}$ Spermatogonia and Spermatocytes.

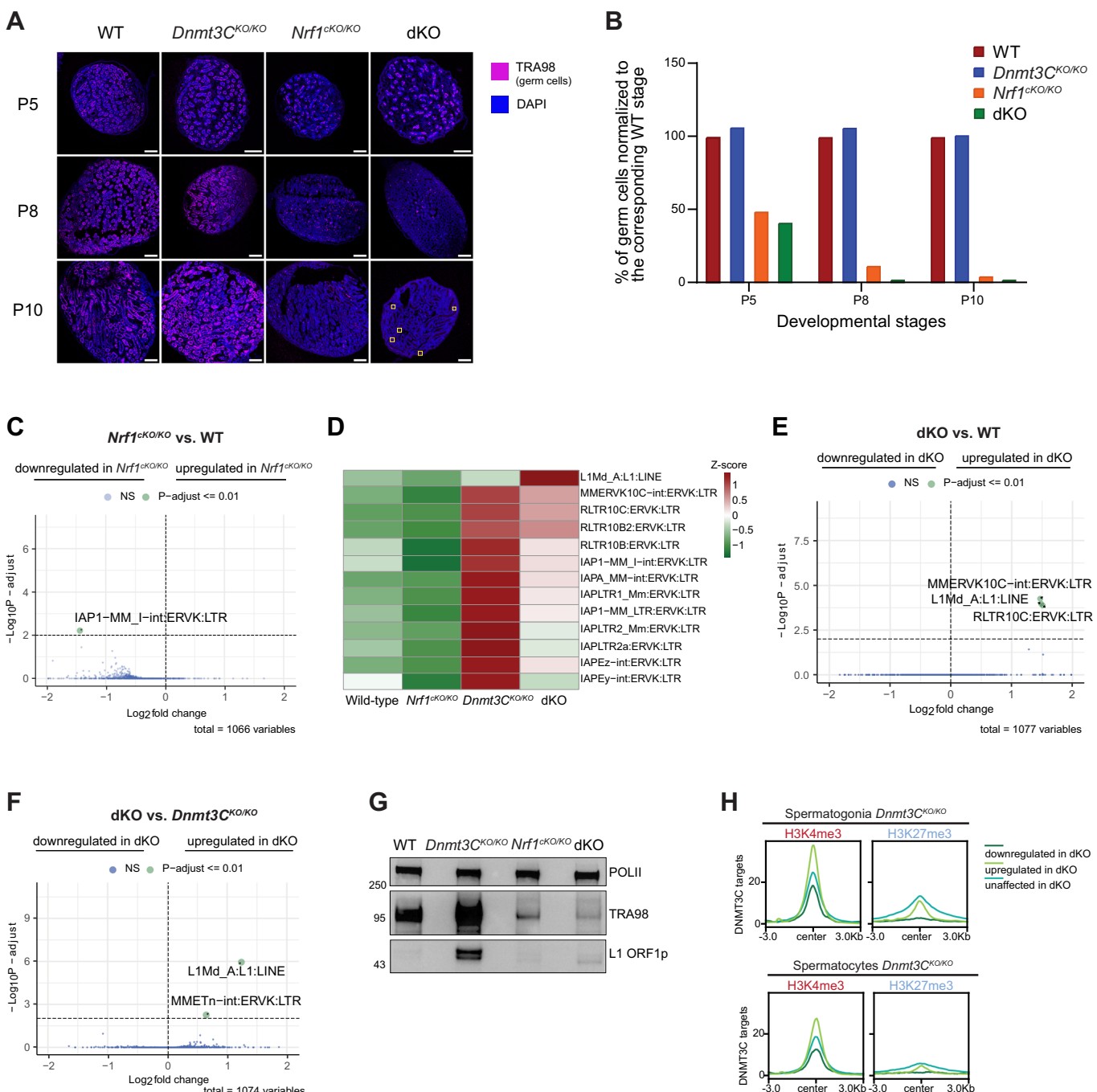

**Figure EV5. Related to Fig. 5.**

(A) Immunostaining of whole-testis cryosections showing TRA98 (germ cell marker) and DAPI at P5, P8 and P10 for wild-type, *Dnmt3C^{KO/KO}*, *Nrf1^{cKO/KO}* and dKO (*Dnmt3C^{KO/KO}*, *Nrf1^{cKO/KO}*) from one biological replicate. Scale bar 200μm. TRA98-marked cells in P10 dKO are marked with yellow boxes for illustration. (B) Bar diagram showing germ cell percentage from the mutants in different stages according to panel (A). TRA98/DAPI counts generated by Fiji ImageJ automated counting from one stage were normalized to TRA98/DAPI counts in the corresponding WT stage from one biological replicate. (C) Volcano plot showing DESeq2 results with log$_2$ FC and -log$_{10}$p-values from Wald test comparing wild-type versus *Nrf1^{cKO/KO}* from three biological replicates. TE names displayed following the pattern TEname:family:TEclass. (D) Heatmaps showing Z-score of TEtranscripts from three biological replicates from differentially expressed TE families in all mutants. TE names displayed following the pattern TEname:family:TEclass. (E) As in (C) for dKO vs. wild-type. (F) As in (C) for dKO vs. *Dnmt3C^{KO/KO}*. (G) Western blot analysis of POL II, TRA98 and LINE1-ORF1p on protein extracts from wild-type, *Dnmt3C^{KO/KO}*, *Nrf1^{cKO/KO}* and dKO testes at P8 from one biological replicate. The blot was developed using femto chemistry. (H) Metaplots showing mean enrichment of H3K4me3 and H3K27me3 CUT&Tag from two biological replicates in wild-type and *Dnmt3C^{KO/KO}* Spg and Spc, centered on DNMT3C targets that were upregulated, downregulated or unaffected in dKO compared to *Nrf1^{cKO/KO}*.

