## [Peer Review File · EMBO Reports]

DNA methylation at retrotransposons protects the germline by preventing NRF1-mediated activation

Joan Barau, Jessica Leismann, Styliani-Eirini Kanta, Ishita Amar, Anna Szczepińska, Monika Mielnicka, Giuseppe Petrosino, Anke Busch, Marion Scheibe, Pengxiang Wang, Yuan Wang, Falk Butter, and Matthieu Boulard

Corresponding author(s): Joan Barau (J.Barau@imb-mainz.de)

Review Timeline:

Submission Date:	8th Jan 25
Editorial Decision:	29th Jan 25
Revision Received:	21st May 25
Editorial Decision:	12th Jun 25
Revision Received:	25th Jun 25
Accepted:	4th Jul 25

Editor: Esther Schnapp

Transaction Report:

Dear Dr. Barau,

Thank you for the transfer of your manuscript to EMBO reports. We have now received the full set of referee reports that is pasted below.

As you will see, the referees acknowledge that the findings are interesting. However, they also have several suggestions for how the study could be strengthened and improved. I think all suggestions are good and should be addressed. Please let me know in case you disagree and we can discuss the exact revision requirements further, also in a video chat, if you like.

I would thus like to invite you to revise your manuscript with the understanding that the referee concerns must be fully addressed and their suggestions taken on board. Please address all referee concerns in a complete point-by-point response. Acceptance of the manuscript will depend on a positive outcome of a second round of review. It is EMBO reports policy to allow a single round of major revision only and acceptance or rejection of the manuscript will therefore depend on the completeness of your responses included in the next, final version of the manuscript.

We realize that it is difficult to revise to a specific deadline. In the interest of protecting the conceptual advance provided by the work, we recommend a revision within 3 months (1st May 2025). Please discuss the revision progress ahead of this time with the editor if you require more time to complete the revisions.

- 1) A data availability section providing access to data deposited in public databases is missing. If you have not deposited any data, please add a sentence to the data availability section that explains that.
- 2) Your manuscript contains statistics and error bars based on $n=2$. Please use scatter blots in these cases. No statistics should be calculated if $n=2$.

5) a complete author checklist, which you can download from our author guidelines <<https://www.embopress.org/page/journal/14693178/authorguide>>. Please insert information in the checklist that is also reflected in the manuscript. The completed author checklist will also be part of the RPF.

6) Please note that all corresponding authors are required to supply an ORCID ID for their name upon submission of a revised manuscript (<<https://orcid.org/>>). Please find instructions on how to link your ORCID ID to your account in our manuscript tracking system in our Author guidelines <<https://www.embopress.org/page/journal/14693178/authorguide#authorshipguidelines>>

7) Before submitting your revision, primary datasets produced in this study need to be deposited in an appropriate public database (see <https://www.embopress.org/page/journal/14693178/authorguide#datadeposition>). Please remember to provide a reviewer password if the datasets are not yet public. The accession numbers and database should be listed in a formal "Data Availability" section placed after Materials & Method (see also <https://www.embopress.org/page/journal/14693178/authorguide#datadeposition>). Please note that the Data Availability Section is restricted to new primary data that are part of this study. * Note - All links should resolve to a page where the data can be accessed. *
If your study has not produced novel datasets, please mention this fact in the Data Availability Section.

12) All Materials and Methods need to be described in the main text using our 'Structured Methods' format, which is required for all research articles. According to this format, the Methods section includes a Reagents and Tools Table (listing key reagents, experimental models, software and relevant equipment and including their sources and relevant identifiers) followed by a Methods and Protocols section describing the methods using a step-by-step protocol format. The aim is to facilitate adoption of the methodologies across labs. More information on how to adhere to this format as well as a downloadable template (.docx) for the Reagents and Tools Table can be found in our author guidelines: <https://www.embopress.org/page/journal/14693178/authorguide#structuredmethods>.

An example of a Method paper with Structured Methods can be found here: <https://www.embopress.org/doi/full/10.1038/s44320-024-00037-6#sec-4>

We would also welcome the submission of cover suggestions, or motifs to be used by our Graphics Illustrator in designing a

cover.

I look forward to seeing a revised form of your manuscript when it is ready.

Referee #1:

In this work, Leismann and colleagues describe the effects of transposable element (TE) reactivation in mouse male germline in the context of DNA methylation sensitivity. It is a follow up work from an initial finding that the mouse-specific Dnmt3c is a methyltransferase that exclusively acts on the germline, and its absence leads to very localised changes on DNA methylation on promoters of some young TEs. The authors set to validate the expression of TEs in that Dnmt3c KO model. Then, look for binding proteins on those regions using proteomics, and identify some transcription factors that might be driving the expression of those demethylated TEs. Finally, they aim to validate Nrf1, a well-known methyl-sensitive TF as an important regulator of TE expression on the germline. Overall the paper is interesting, makes relevant discoveries in the field of TE regulation in the germline.

This said, there are several aspects that would require further clarification. I have some major points below:

- Mouse germline methylation is quite dynamic. Beyond the TEs methylated by Dnmt3c, there is a whole demethylation wave, also some demethylation during meiosis, see PMID: 29618374 or PMID: 37723297. Since methylation dynamics are quite important in the context of this work, this should be made way more explicit for the reader, since several developmental stages of germ cells are taken, and it is hard to reconcile what we're looking at with what is known about DNA methylation dynamics on this system. Perhaps the samples are no longer available, but having some sort of antibody validation of methylation on those different samples could give a sense on global methylation levels. If there is global demethylation, then, the Nrf1 binding landscape is very likely to change, way beyond the Dnmt3C target sites.
- While the analysis of RNA-seq is done using a specialised pipeline for TEs (TEtranscripts/TElocal), the CUT&Tag is not. Random assignment of multimapping reads has been deemed a decent compromise for this kind of data, and that's what the authors have used. But then, locus specificity and classification of various TEs in several histone enrichments might be affected by that. Perhaps using uniquely mapping reads on the borders of the TEs (these are young TEs, so I don't expect much unique mappability within the TE) would be a nice way to validate their method. Given that the activity seems to be on the promoters of these TEs, focusing on borders should also reveal some level of enrichment.
- The proteomics experiments do not show many of the classic generic methylation (or lack of methylation) readers previously reported in several papers (PMID: 23434322). While UHRF1 is there, there is no MBD or CXXC containing protein. Is this perhaps because these genes are not expressed in these cells? Since there is RNA-seq, perhaps having a look would be interesting, to grasp the level of sensitivity of these pull downs (these should appear in all methylation vs not methylation comparisons, including the scrambled control).
- Figure 3C. This heatmap shows some background binding on the WT. This could be an artifact of CUT&Tag, that has some background preferences associated to Tn5. A way to test this is to use `bamCoverage` to generate a track enriched / subtracted from IgG control, or show the raw mapping of the input track on those regions. Otherwise, the difference in binding is very subtle, and not much can be concluded (is really Nrf1 not binding in WT?). Then, a more quantitative approach would also be recommended, e.g. counting reads on all these regions in both samples and running them through DEseq2 / DiffBind.
- It is said that the Nrf1 cKO/KO shows very few differentially expressed protein coding genes. Then, the number is ~2000 (between up and downregulated genes). This is not "few" to me. Perhaps some extra elaboration on that result would be nice.
- The whole Nrf1 KO is a really tricky experiment to interpret, since this is a very essential TF. I do believe that TEs are probably using this TF to amplify their expression, but expressing more nuance would be good, especially when the CUT&Tag (e.g. figure 3C) doesn't show a very clear pattern. Also, some reconciliation on the broad (potential) binding of Nrf1 to Dnmt3c targets and lack of LINE response could be better developed.

Then, I have some minor comments:

In the abstract it reads "Conditional germline knockout of Nrf1 in the absence of DNA methylation rescued the patterns of reactivation of the most mutagenic retrotransposon in mice - Intracisternal A-particle or IAP ". Similar sentences are throughout

the manuscript. Confusing sentence, since "rescuing expression" kind of suggests that they are expressed, but this would rescue the silencing, no?

Figure 1D - what are the downregulated TEs in the Dnmt3C KO? Why they are not highlighted?

Why using scare quotes around "developmental-like expression pattern"? Perhaps explain what you mean a bit better or cite the reference for this expression, I don't feel particularly triggered by this expression, so the quotes make me feel there might be something wrong there.

Not sure why the mechanistic model explaining these findings is in Supplementary Figure 10G. Even if some of the current evidence is hard to interpret due to technical limitations, this belongs to main text as a working hypothesis of current data. If this turns out not to be true in the future, so be it, but I think it is a nice discussion interpretation of the results.

Referee #2:

PIWI interacting RNAs (piRNAs) repress transposable elements in the animal germline. One key effector of this pathway is the de novo DNA methyltransferase DNMT3C. The PI has previously discovered DNMT3C and shown that the protein is essential for mouse male fertility and transposon silencing. In this article, Leismann and Kanta & al. evaluate the aftermath of hypomethylation following DNMT3C deletion, from the perspective of the epigenome. They uncover that, as predicted, transposons are reactivated despite the presence of a functional piRNA pathway. They track this reactivation across mouse spermatogenesis and reveal distinct expression profiles for these transcripts, likely governed by equally distinct chromatin marks. The authors use CUT&Tag to reveal the landscape of these marks. In the absence of DNA methylation, the authors reveal a slight enrichment of transcription factors NRF1 and CREB1 on activated transposons. To validate the role of NRF1 in transcriptional activation, the authors employ Nrf1 conditional knockout mice models that when combined with Dnmt3c knockout, reestablish repression of demethylated promoters. All in all, the authors emphasize the role of Nrf1 in transcription of transposon elements and reveal yet another hierarchical organization of transposon activation. This is one of the few studies (apart from previously described CREB1) that characterizes transcription factors that drive expression of transposon transcripts. I am very enthusiastic about this study and the value that it brings to the field of transposon biology and piRNA pathway. I am happy to support its publication after minor edits as listed below.

Comments:

The scaling in Supp Fig 1A is not obvious and values do not add up to 100.

The manuscript is a bit too long. It can be shortened a bit through rephrasing and shortening the results section.

It would be interesting to know if NRF1 and CREB1 are occluded from performing their transcription activation roles via additional mechanisms (not limited to methylated loci) such as post-translational modifications or interactions with inhibitors. Please discuss.

The enrichment of NRF1 on certain loci following loss of methylation is minor. The authors use CUT&Tag to map NRF1 binding site. While the novel approach provides excellent results for mapping chromatin marks in high resolution, the method is still insufficient to map transcription factor binding at the same resolution. This should be acknowledged in the discussion, not only for CREB1, but also for NRF1.

Referee #3:

In this manuscript (DNA methylation at retrotransposons protects the germline by preventing NRF1-mediated activation), Leismann et al. characterize the impact of Dnmt3C KO on the chromatin landscape of developing germ cells and the mechanisms that lead to selective activation of transposable elements.

In summary, the authors document persistent L1-ORF1 expression in the Dnmt3C KO through P30 (immunofluorescence & western data Figure 1), extending the time course of P20 data previously published (Barau et al 2016 Fig 2E). They add RNA-seq data from FACS-isolated spermatogonia and spermatocytes consistent with previous bulk RNA-seq data from the Dnmt3C KO (Barau et al 2016 Fig 2C). FACS-isolated germ cells are used for important CUT&Tag experiments to explore the effect of Dnmt3C KO on bivalent chromatin marks and categorize DNMT3C targets by chromatin signature. The authors proceed using elegant pulldown experiments to identify NRF1 as a candidate methylation-sensitive activator. They subsequently provide important evidence linking NRF1 to transcriptional activation of unmethylated retrotransposons by profiling NRF1 occupancy across germ cell stages and exploring the impact on TE expression following NRF1 germ-cell depletion.

Overall, the manuscript presents important data contributing to understanding the mechanisms of retrotransposon silencing in the mammalian germ line that is likely to be of broad interest across the fields of chromatin biology, transcriptional regulation, genomics, development and evolution. The data is clearly presented and includes important observations that required time-consuming experiments to generate the required germ-cell deletion line and considerable expertise to complete the CUT&Tag experiments on sorted germ cell populations. I provide the following suggestions for consideration to strengthen the manuscript

in a few areas:

1. CUT&Tag analysis methods: You indicate that BAM files were merged for duplicate samples. I expect that differences between biological samples could add quite a bit of noise. Can you explain why you chose this pipeline over processing the data separately and identifying replicated peaks and significant differences between genotype? This is relevant because some of the observed differences are marginal and not quantified. I would like to see a more robust statistical comparison and/or transparency of the data for each replicate. This would address concerns that peak calling of weak peaks might impact classification and provide more confidence on the NRF1 occupancy data.
2. The CUT&Tag data presentation (Figure 2/S3): Conclusions from this data are dependent on comparisons between genotypes and also between time points and peak classification categories. I found it difficult to follow and the few conclusions from this section did not come through clearly without several re-readings. Could the genotypes & comparisons be clarified for the reader with additional labels in the figures and/or regrouping of figure panels? For example, the text statement "In Dnmt3CKO/KO Spg, we observed a marked increase of H3K27me3 when compared to E15.5 prospermatogonia, particularly in the likely- bivalent and H3K27me3-marked categories (Figure 2B)". Figure 2B does not appear to have any prospermatogonia data so I am not sure what I am comparing the 2B data to. Presumably data in 3SB? But genotype of the 3SB data is not provided in either the panel or the legend.
3. The conclusions from the CUT&Tag experiments seem to rely on quantitative comparisons that are sometimes subtle and not always convincing in the aggregate heatmaps. For example, the conclusion "in Dnmt3CKO/KO spermatocytes (Spc), H3K27me3 becomes markedly reduced while active H3K4me3 is retained at DNMT3C targets". From the heat maps provided in 2A, it appears both marks are reduced in KO Spc compared to KO Spg (~30% reduction for H3K4me3 compared to ~50% reduction H3K27me3 in the metaplots). Can you provide further explanation of your conclusion that the H3K4me3 is more strongly retained? Was this a subjective call based on the heat map visualization, or a more quantitative comparison that I missed in the methods?
4. Similarly, I would like to see further support for some conclusions surrounding Figure 2C. For example, the authors state "Interestingly, the bivalent category displayed a more bimodal distribution" and that they "follow a distinct expression pattern". Both the bivalent & H3K4me3 marked categories appear bimodal in the aggregate violin plot - what is the evidence that the bivalent are "more bimodal" or "distinct" from the H3K3me-marked?

Additional minor typos/comments:

1. Please include page numbers and/or line numbers for future submissions to help reviewers provide comments.
2. Pg4. Typo: "in response to the loss of promoter DNA methylation (Supp. Figure 2A-C)" Should this reference Figure 1C-E instead?
3. Supp Figure 1A: It might be appropriate to reference the Ernst et al data that is presented in Supp Fig in the main text, not just the supplemental figure legend
4. Supp Figure 1B: Is duplication of this figure is needed (Fig 1A contrast adjusted & Supp Fig 1B raw image)? This data extends the time course, but presents nothing new compared to persistent expression of L1-ORF1 immunostaining data previously published for P20 (Barau et al 2016 Fig 1E).
5. Supplemental Figure 2A: Could a plot of the KO Spg population be shown (similar to the Spc in panel 2C)? The profile in the KO is clearly quite different for Spc (this limitation is acknowledged by the authors due to KO pachytene arrest). It would be nice to see visual confirmation that the Spg populations are more similar.
6. Pg 6: "This suggests that either IAPs and MMERVK10C are transcribed from a smaller number of elements (Rebollo et al, 2020), but have stronger promoters.": This either/but statement is not clear.
7. Figure 3 legend: "Spg and Spcoverlapping with" is missing a space.

Submission of revised manuscript to EMBO Reports

Dear Dr. Schnapp,

We are pleased to submit the revised version of our manuscript entitled “*DNA methylation at retrotransposons protects the germline by preventing NRF1-mediated activation.*” In this version, we have thoroughly addressed all reviewer comments, either by implementing the suggested changes—including a re-analysis of our data using more stringent criteria that retain only uniquely mapped reads and performing a differential enrichment analysis, which further reinforces our original conclusions—or by providing a detailed explanation in cases where specific experimental suggestions could not be implemented. We also welcome a new co-author, Anna Szczepińska, who contributed to these additional analyses, with the full agreement of all authors.

A version of the manuscript with highlighted changes and a point-by-point response to each reviewer comment are included with this submission.

We thank you and the reviewers for your constructive feedback, which has helped us substantially improve the clarity and robustness of the manuscript.

Sincerely,

Joan Barau

Response to Reviewers

Referee #1:

In this work, Leismann and colleagues describe the effects of transposable element (TE) reactivation in mouse male germline in the context of DNA methylation sensitivity. It is a follow up work from an initial finding that the mouse-specific Dnmt3c is a methyltransferase that exclusively acts on the germline, and its absence leads to very localised changes on DNA methylation on promoters of some young TEs. The authors set to validate the expression of TEs in that Dnmt3c KO model. Then, look for binding proteins on those regions using proteomics, and identify some transcription factors that might be driving the expression of those demethylated TEs. Finally, they aim to validate Nrf1, a well-known methyl-sensitive TF as an important regulator of TE expression on the germline. Overall the paper is interesting, makes relevant discoveries in the field of TE regulation in the germline. This said, there are several aspects that would require further clarification. I have some major points below:

1. Mouse germline methylation is quite dynamic. Beyond the TEs methylated by Dnmt3c, there is a whole demethylation wave, also some demethylation during meiosis, see PMID: 29618374 or PMID: 37723297. Since methylation dynamics are quite important in the context of this work, this should be made way more explicit for the reader, since several developmental stages of germ cells are taken, and it is hard to reconcile what we're looking at with what is known about DNA methylation dynamics on this system. Perhaps the samples are no longer available, but having some sort of antibody validation of methylation on those different samples could give a sense on global methylation levels. If there is global demethylation, then, the Nrf1 binding landscape is very likely to change, way beyond the Dnmt3C target sites.

We thank the reviewer for raising this important point. Indeed, we agree that the dynamic nature of DNA methylation in the germline adds complexity to the interpretation of stage-specific TE regulation. To assist the reader in navigating the developmental stages examined in this study, and to clarify the expected methylation context, we have prepared an additional version of Figure EV1 (see below) summarizing the developmental stages analyzed, the predominant germ cell types present at each stage, and the expected DNA methylation levels based on published data. We would be happy to include this figure in the manuscript if the reviewer finds it helpful.

Regarding the broader dynamics of DNA methylation: We also fully acknowledge the well-established, genome-wide wave of demethylation and subsequent remethylation in embryonic prospermatogonia (PMID: 3428592, 23219530, 37508536). However, the evidence supporting a second global wave of DNA demethylation during meiosis, remains controversial and scarce. To our knowledge, the most comprehensive evidence against any global DNA methylation reprogramming event in postnatal spermatogenesis comes from Fanourgakis et al (PMID: 39774947), who conditionally deleted both *Dnmt3a* and *Dnmt3b* in spermatogonia, thus removing all active de novo DNA methyltransferases activity in postnatal germ cell development in the mouse (mind DNMT3C is present only in embryonic prospermatogonia). Then, they assessed the DNA methylation levels at single base pair resolution by Enzymatic Methyl-seq (EM-seq) in FACS sorted sperm, profiling 99.0% of all genomic CpGs at a coverage of 6.5X. Despite the complete absence of de novo methylation for entire 35 days of germ cell development from spermatogonia, through meiotic spermatocytes and

round spermatids to sperm, they observed only a modest global methylation decrease of 5.2% (81.7% to 76.5%) in sperm. Importantly, repeats were underrepresented among the hypomethylated regions, indicating that repetitive elements retained their methylation throughout postnatal germ cell development. In our opinion, this experiment provides strong genetic and epigenomic evidence that a global demethylation event during meiosis is unlikely, and that repetitive elements, including TEs, are not major targets of postnatal methylation loss during spermatogenesis.

While we cannot directly exclude subtle changes in methylation at specific loci, we emphasize that our study focuses on DNMT3C target TEs, for which DNA methylation is lost earlier during embryonic development (Barau et al. 2016). We observed increased NRF1 binding at these elements in spermatocytes of *Dnmt3c* KO, consistent with their persistent hypomethylation (Figure 3C). Thus, our main conclusions about NRF1 binding to TEs are independent of additional methylation dynamics not specific to TEs (and controlled for in our WT samples).

We agree with the reviewer that a direct readout of methylation levels across stages would be informative. However, 5mC immunostaining lacks the resolution and consistency required to reliably compare methylation levels across germ cell subtypes, as shown for instance in PMID: 31624244 (Fig. 4C as an example in spermatogenesis – where the staining in sections showing naïve spermatogonia > differentiating spermatogonia > spermatocytes > spermatids is very heterogeneous, far beyond the variation shown by WGBS for some of these stages). To effectively measure DNA methylation in our experimental setup, we would have to obtain more animals, sort germ cell stages and perform WGBS, which would essentially repeat the effort of several already existing high-quality studies (e.g. PMID: 33998651, PMID: 27856912, PMID: 29728652 and a few others).

Finally, we now included a brief clarification in the discussion (see line 510-513, page 12) to contextualize our findings with respect to DNA methylation dynamics and to clearly state that our interpretation is limited to DNMT3C targets and does not generalize to the genome-wide behavior of NRF1.

2. While the analysis of RNA-seq is done using a specialised pipeline for TEs

(TEtranscripts/TElocal), the CUT&Tag is not. Random assignment of multimapping reads has been deemed a decent compromise for this kind of data, and that's what the authors have used. But then, locus specificity and classification of various TEs in several histone enrichments might be affected by that. Perhaps using uniquely mapping reads on the borders of the TEs (these are young TEs, so I don't expect much unique mappability within the TE) would be a nice way to validate their method. Given that the activity seems to be on the promoters of these TEs, focusing on borders should also reveal some level of enrichment.

We thank the reviewer for this thoughtful and constructive suggestion. In response, we re-analyzed the CUT&Tag histone modification data using only uniquely mapping reads. This approach indeed allowed us to more precisely resolve the chromatin features at TE promoters and confirmed the presence of specific histone mark patterns across TE families. We now include both the original analysis (based on randomly assigned multimappers) and the new uniquely mapped read analysis in the manuscript to provide a more comprehensive and robust view of the chromatin states. These results are presented in Figure 2, Figure EV2 and Appendix Figure S2, and the main text and methods have been updated accordingly.

3. The proteomics experiments do not show many of the classic generic methylation (or lack of methylation) readers previously reported in several papers (PMID: 23434322). While UHRF1 is there, there is no MBD or CXXC containing protein. Is this perhaps because these genes are not expressed in these cells? Since there is RNA-seq, perhaps having a look would be interesting, to grasp the level of sensitivity of these pull downs (these should appear in all methylation vs not methylation comparisons, including the scrambled control).

We appreciate this insightful suggestion. To address this point, we examined the expression of known methylation readers, including MBD- and CXXC-domain containing proteins, using our RNA-seq data from spermatogonia and spermatocytes. We found that many of the canonical methylation readers listed in PMID: 23434322 are indeed lowly expressed in these cells. This likely contributes to their limited representation in our methyl-DNA pulldown proteomics. Nonetheless, several DNA methylation readers were enriched in our pulldown experiments, including UHRF1, MBD1, MBD2, and MECP2, particularly on methylated probes compared to unmethylated. However, the enrichment was not found to be significant. We are happy to share the detailed data with the reviewer hereafter. Below, we plotted the expression of methyl-binding proteins in wild-type Spg and Spc on the left and their differential enrichment on unmethylated vs. methylated probes on the right:

4. Figure 3C. This heatmap shows some background binding on the WT. This could be an artifact of CUT&Tag, that has some background preferences associated to Tn5. A way to test this is to use `bamCoverage` to generate a track enriched / subtracted from IgG control, or show the raw mapping of the input track on those regions. Otherwise, the difference in binding is very subtle, and not much can be concluded (is really Nrf1 not binding in WT?). Then, a more quantitative approach would also be recommended, e.g. counting reads on all these regions in both samples and running them through DEseq2 / DiffBind.

We thank the reviewer for this helpful suggestion. To more rigorously assess NRF1 binding, we have now incorporated additional analyses that strengthen our conclusions. Specifically, we included coverage tracks for the IgG control in Figure EV4B and EV4C, allowing a clearer assessment of background signal.

Moreover, we performed a differential enrichment analysis using RepEnrich2 followed by EdgeR, comparing NRF1 binding in *Dnmt3C* mutants vs wild-type, and also NRF1 vs IgG in both genotypes. These analyses revealed that NRF1 binding is significantly higher in *Dnmt3C* mutants compared to wild-type, and this enrichment is also above IgG background in the mutant samples. These new results have been integrated into the manuscript: the mutant vs wild-type analysis is presented in main Figure 4B,C, and the NRF1 vs IgG comparisons are included in Appendix Figure S7.

We also acknowledge in the revised Discussion that TF profiling in low-input primary cells remains technically challenging. Among the methods tested (CUT&Run and ULI-ChIP), CUT&Tag proved to be the most effective under our experimental conditions, even if it still presents limitations in sensitivity and resolution compared to its performance on chromatin marks.

5. It is said that the Nrf1 cKO/KO shows very few differentially expressed protein coding genes.

Then, the number is ~2000 (between up and downregulated genes). This is not "few" to me. Perhaps some extra elaboration on that result would be nice.

We thank the reviewer for pointing this out. Indeed, the original wording was misleading. We have now revised the text to clarify that a substantial number of protein-coding genes are differentially expressed in *Nrf1cKO/KO* testes compared to wild-type.

6. The whole Nrf1 KO is a really tricky experiment to interpret, since this is a very essential TF. I do believe that TEs are probably using this TF to amplify their expression, but expressing more nuance would be good, especially when the CUT&Tag (e.g. figure 3C) doesn't show a very clear pattern. Also, some reconciliation on the broad (potential) binding of Nrf1 to Dnmt3c targets and lack of LINE response could be better developed.

We appreciate this thoughtful comment and fully agree that interpreting the *Nrf1* cKO experiment requires caution, given the essential role of NRF1 in multiple biological processes. In response, we have revised the Discussion to more clearly acknowledge the complexity of NRF1's function and to add nuance to our interpretation of the data. Specifically, we now discuss that while NRF1 appears to bind broadly to many DNMT3C targets, its impact on TE expression is selective (for reasons unknown). We also discuss the apparent lack of response among LINE elements, which may reflect additional layers of regulation and additive effects or compensatory mechanisms, such as the action of other transcription factors, chromatin context and we explicitly mention the possibility of a heterogeneous response among different L1 subfamilies (see lines 528-547, page 12).

In these revised sections, we also explore the possibility that a transcriptional L1 response could still be occurring but may have escaped detection in our bulk RNA-seq data due to technical limitations. To investigate this, we quantified L1-ORF1 protein levels in mutant germ cells using immunofluorescence (Appendix Figure S8D). The results support the idea that overall L1 protein levels may not mirror the changes observed in transcript levels for individual subfamilies, possibly due to the downregulation of other L1 subfamilies. We would be happy to retain this interpretation in the Discussion if the reviewer finds it a relevant addition.

Lastly, we also point out that the relatively subtle patterns observed in CUT&Tag (old Figure 3C, new: Figure 4A) are consistent with the idea that NRF1 activity might not only be modulated by DNA binding but also by co-factor availability and chromatin context. The revised manuscript presents a more balanced interpretation and highlights the need for further mechanistic exploration.

Then, I have some minor comments:

7. In the abstract it reads "Conditional germline knockout of *Nrf1* in the absence of DNA methylation rescued the patterns of reactivation of the most mutagenic retrotransposon in mice - Intracisternal A-particle or IAP ". Similar sentences are throughout the manuscript. Confusing sentence, since "rescuing expression" kind of suggests that they are expressed, but this would rescue the silencing, no?

We thank the reviewer for this helpful suggestion. We agree that the phrasing was ambiguous. We have revised all relevant instances throughout the manuscript to clarify this point, now using the phrasing "rescuing the silencing of IAPs" to more accurately reflect our findings.

8. Figure 1D - what are the downregulated TEs in the Dnmt3C KO? Why they are not highlighted? Why using scare quotes around "developmental-like expression pattern"? Perhaps explain what you mean a bit better or cite the reference for this expression, I don't feel particularly triggered by this expression, so the quotes make me feel there might be something wrong there.

We thank the reviewer for this constructive comment. Initially, we omitted the downregulated TEs from the figure due to space constraints, as there were too many to display clearly. However, we agree that it is informative to include them. We have now updated Figure 1D to highlight the top downregulated TEs in *Dnmt3C KO* spermatocytes. We also provide in the result section the likely explanation for these - that additional cells encompassing later stages of meiosis are also present in samples from WT but absent in *Dnmt3C KO* due to its well known and documented developmental arrest of spermatogenesis (at pachytene stage of meiosis). Downregulated TEs in WT reflect the older, mostly retrotransposition inactive TEs that are part of the transcriptional landscape of these cells.

Regarding the phrase “developmental-like expression pattern”, we acknowledge that the phrasing and the quotation marks may have been confusing. We intended to convey that the expression of the TE families was heterogeneous across developmental stages. To improve clarity, we have rephrased this now as “varied expression pattern” throughout the manuscript.

9. Not sure why the mechanistic model explaining these findings is in Supplementary Figure 10G. Even if some of the current evidence is hard to interpret due to technical limitations, this belongs to main text as a working hypothesis of current data. If this turns out not to be true in the future, so be it, but I think it is a nice discussion interpretation of the results.

We fully agree with the reviewer that the mechanistic model helps to synthesize our findings. In response, we have improved the initial model and moved it from Supplementary Figure 10G into the main figures (now Figure 6) to better highlight our proposed interpretation.

Referee #2:

PIWI interacting RNAs (piRNAs) repress transposable elements in the animal germline. One key effector of this pathway is the de novo DNA methyltransferase DNMT3C. The PI has previously discovered DNMT3C and shown that the protein is essential for mouse male fertility and transposon silencing. In this article, Leismann and Kanta & al. evaluate the aftermath of hypomethylation following DNMT3C deletion, from the perspective of the epigenome. They uncover that, as predicted, transposons are reactivated despite the presence of a functional piRNA pathway. They track this reactivation across mouse spermatogenesis and reveal distinct expression profiles for these transcripts, likely governed by equally distinct chromatin marks. The authors use CUT&Tag to reveal the landscape of these marks. In the absence of DNA methylation, the authors reveal a slight enrichment of transcription factors NRF1 and CREB1 on activated transposons. To validate the role of NRF1 in transcriptional activation, the authors employ Nrf1 conditional knockout mice models that when combined with *Dnmt3c* knockout, reestablish repression of demethylated promoters. All in all, the authors emphasize the role of Nrf1 in transcription of transposon elements and reveal yet another hierarchical organization of transposon activation. This is one of the few studies (apart from previously described CREB1) that characterizes transcription factors that drive expression of transposon transcripts. I am very enthusiastic about this study and the value that it brings to the field of transposon biology and piRNA pathway. I am happy to support its publication after minor edits as listed below.

Comments:

1. The scaling in Supp Fig 1A is not obvious and values do not add up to 100.

We thank the reviewer for pointing this out. As the absolute number of germ cells at each developmental stage has not been published, the figure was intended as a schematic estimation to guide the reader through the relative contribution of each germ cell type in the composition of whole gonads in each day of postnatal development. Nevertheless, we have revised and improved the scaling and visual clarity of Supplementary Figure 1A now as Figure EV1A to avoid confusion.

2. The manuscript is a bit too long. It can be shortened a bit through rephrasing and shortening the results section.

We agree with the reviewer's suggestion and have revised the manuscript to improve clarity and conciseness. In particular, we rephrased and shortened several parts of the Results section (specifically on figure 2) while preserving the key findings and interpretations.

3. It would be interesting to know if NRF1 and CREB1 are occluded from performing their transcription activation roles via additional mechanisms (not limited to methylated loci) such as post-translational modifications or interactions with inhibitors. Please discuss.

We agree with the reviewer that additional layers of regulation beyond DNA methylation could influence the transcriptional activity of NRF1 and CREB1. While our data certainly point in this direction, we do not have means to experimentally test in germ cells the observed correlations. In the revised Discussion, we now address the possibility that NRF1 may also be occluded from binding or activation due to repressive histone modifications (e.g., H3K27me3), or post-translational modifications such as phosphorylation. These mechanisms may contribute to the selective activation or repression of transposable elements and gene targets during germ cell development.

4. The enrichment of NRF1 on certain loci following loss of methylation is minor. The authors use CUT&Tag to map NRF1 binding site. While the novel approach provides excellent results for mapping chromatin marks in high resolution, the method is still insufficient to map transcription factor binding at the same resolution. This should be acknowledged in the discussion, not only for CREB1, but also for NRF1.

We thank the reviewer for this important point. We now acknowledge in the revised Discussion that while CUT&Tag is a powerful technique for profiling histone modifications with high resolution and low background, its application to TF mapping has certain limitations, especially in low-input primary cells. In our hands, CUT&Tag proved to be the most effective approach (after testing other chromatin profiling techniques as CUT&Run and ULI-ChIP) under these experimental conditions, even if still suboptimal for precise TF binding site identification. We have added this nuance to the discussion to better contextualize our findings and the interpretation of CREB1 and NRF1 enrichment signals.

To address the concern about the subtle NRF1 enrichment, we have now performed a differential enrichment analysis using RepEnrich2. This analysis revealed that despite the modest overall signal, NRF1 binding is significantly enriched at specific transposon families in the absence of DNA methylation. These results help to support our conclusions and are now included in Figures 4B, 4C, 4E, and Appendix Figure S7.

Referee #3:

In this manuscript (DNA methylation at retrotransposons protects the germline by preventing NRF1-mediated activation), Leismann et al. characterize the impact of Dnmt3C KO on the chromatin landscape of developing germ cells and the mechanisms that lead to selective activation of transposable elements.

In summary, the authors document persistent L1-ORF1 expression in the Dnmt3C KO through P30 (immunofluorescence & western data Figure 1), extending the time course of P20 data previously published (Barau et al 2016 Fig 2E). They add RNA-seq data from FACS-isolated spermatogonia and spermatocytes consistent with previous bulk RNA-seq data from the Dnmt3C KO (Barau et al 2016 Fig 2C). FACS-isolated germ cells are used for important CUT&Tag experiments to explore the effect of Dnmt3C KO on bivalent chromatin marks and categorize DNMT3C targets by chromatin signature. The authors proceed using elegant pulldown experiments to identify NRF1 as a candidate methylation-sensitive activator. They subsequently provide important evidence linking NRF1 to transcriptional activation of unmethylated retrotransposons by profiling NRF1 occupancy across germ cell stages and exploring the impact on TE expression following NRF1 germ-cell depletion. Overall, the manuscript presents important data contributing to understanding the mechanisms of retrotransposon silencing in the mammalian germ line that is likely to be of broad interest across the fields of chromatin biology, transcriptional regulation, genomics, development and evolution. The data is clearly presented and includes important observations that required time-consuming experiments to generate the required germ-cell deletion line and considerable expertise to complete the CUT&Tag experiments on sorted germ cell populations. I provide the following suggestions for consideration to strengthen the manuscript in a few areas:

1. CUT&Tag analysis methods: You indicate that BAM files were merged for duplicate samples. I expect that differences between biological samples could add quite a bit of noise. Can you explain why you chose this pipeline over processing the data separately and identifying replicated peaks and significant differences between genotype? This is relevant because some of the observed differences are marginal and not quantified. I would like to see a more robust statistical comparison and/or transparency of the data for each replicate. This would address concerns that peak calling of weak peaks might impact classification and provide more confidence on the NRF1 occupancy data.

We thank the reviewer for raising this important point. To address these concerns, we have now included the individual replicate profiles of NRF1 binding in Figure EV4B for complete transparency. In addition, we performed a statistical analysis of NRF1 enrichment using RepEnrich2 followed by EdgeR, which quantifies log-fold changes in NRF1 binding at different IAP and L1 subfamilies across genotypes. This analysis now allows us to rigorously assess differential NRF1 occupancy and its significance, and the results are presented in revised Figure 4B, 4C, and 4E, as well as in Appendix Figure S7. These additions strengthen the robustness of our CUT&Tag data and support our conclusions about NRF1 binding dynamics.

2. The CUT&Tag data presentation (Figure 2/S3): Conclusions from this data are dependent on comparisons between genotypes and also between time points and peak classification categories. I found it difficult to follow and the few conclusions from this section did not come through clearly without several re-readings. Could the genotypes & comparisons be clarified for the reader with additional labels in the figures and/or regrouping of figure panels? For example, the text statement "In Dnmt3CKO/KO Spg, we observed a marked increase of H3K27me3 when compared to E15.5 prospermatogonia, particularly in the likely- bivalent and H3K27me3-marked categories (Figure 2B)". Figure 2B does not appear to have any prospermatogonia data so I am not sure what I am comparing the 2B data to. Presumably data in 3SB? But genotype of the 3SB data is not provided in either the panel or the legend.

We thank the reviewer for this valuable feedback. To improve clarity, we have revised both the figure layout and the corresponding text. Specifically, we have restructured the figures to follow the logical order in the main text and added clear labels indicating the genotypes, developmental stages, and chromatin categories in each panel. In addition, we rephrased the relevant results section to improve readability. We also corrected the figure references to avoid confusion, the comparison involving prospermatogonia is now correctly referenced to the appropriate appendix panel (Appendix Figure S2B,C). We hope these changes make the figure presentation and interpretation significantly clearer.

3. The conclusions from the CUT&Tag experiments seem to rely on quantitative comparisons that are sometimes subtle and not always convincing in the aggregate heatmaps. For example, the conclusion "in Dnmt3CKO/KO spermatocytes (Spc), H3K27me3 becomes markedly reduced while active H3K4me3 is retained at DNMT3C targets". From the heat maps provided in 2A, it appears both marks are reduced in KO Spc compared to KO Spg (~30% reduction for H3K4me3 compared to ~50% reduction H3K27me3 in the metaplots). Can you provide further explanation of your conclusion that the H3K4me3 is more strongly retained? Was this a subjective call based on the heat map visualization, or a more quantitative comparison that I missed in the methods?

We appreciate the reviewer's careful observation and agree that the conclusion regarding H3K4me3 retention was not sufficiently supported by quantitative analysis. While there is indeed a reduction in both histone marks in *Dnmt3C* mutant spermatocytes, the differential retention was not quantitatively assessed and the distinction may be misleading when based only on visual inspection of heatmaps. To address this, we have revised the text to remove the claim about H3K4me3 retention and instead focus on the clearer observation of H3K27me3 loss and gain of H3K27ac in spermatocytes. We believe this adjustment reflects the data more accurately.

4. Similarly, I would like to see further support for some conclusions surrounding Figure 2C. For example, the authors state "Interestingly, the bivalent category displayed a more bimodal distribution" and that they "follow a distinct expression pattern". Both the bivalent & H3K4me3 marked categories appear bimodal in the aggregate violin plot - what is the evidence that the bivalent are "more bimodal" or "distinct" from the H3K3me-marked?

We thank the reviewer for pointing this out. We agree that the previous statement about the bivalent category displaying a "more bimodal" or "distinct" pattern was not sufficiently supported by the data. We have removed this claim from the manuscript and instead now focus on statistically significant differences in expression between the chromatin categories. To further improve our analysis, we additionally plotted the violin plots using chromatin categories derived from uniquely mapped data only. This approach allows for a more accurate comparison by excluding TEs with ambiguous mapping or no detectable expression, thereby improving the interpretability of expression heterogeneity across chromatin states (Figure EV2C, Appendix Figure S4).

Additional minor typos/comments:

5. Please include page numbers and/or line numbers for future submissions to help reviewers provide comments.

We appreciate this suggestion and will ensure that future submissions include both page and line numbers to facilitate the review process.

6. Pg4. Typo: "in response to the loss of promoter DNA methylation (Supp. Figure 2A-C)" Should this reference Figure 1C-E instead?

Thank you for spotting this. The reference was indeed incorrect and has now been updated to Figure 1C-E.

7. Supp Figure 1A: It might be appropriate to reference the Ernst et al data that is presented in Supp Fig in the main text, not just the supplemental figure legend

We agree and have now included a citation and reference to the Ernst et al. dataset in the main text, where the data from Supp. Fig. 1A (now Figure EV1A) is first discussed.

8. Supp Figure 1B: Is duplication of this figure is needed (Fig 1A contrast adjusted & Supp Fig 1B raw image)? This data extends the time course, but presents nothing new compared to persistent expression of L1-ORF1 immunostaining data previously published for P20 (Barau et al 2016 Fig 1E).

We agree that this figure does not add new information. We have removed Supp. Fig. 1B from the revised manuscript.

9. Supplemental Figure 2A: Could a plot of the KO Spg population be shown (similar to the Spc in panel 2C)? The profile in the KO is clearly quite different for Spc (this limitation is acknowledged by the authors due to KO pachytene arrest). It would be nice to see visual confirmation that the Spg populations are more similar.

We agree this would provide helpful context. We have now included a comparable plot for the KO Spg population in the revised Appendix Figure S1A-B, showing a similar Spg population between WT and KO.

10. Pg 6: "This suggests that either IAPs and MMERVK10C are transcribed from a smaller number of elements (Rebollo et al, 2020), but have stronger promoters.": This either/but statement is not clear.

Thank you for catching this. We have revised the sentence.

11. Figure 3 legend: "Spg and Spcoverlapping with" is missing a space.

Thank you for your attention to detail. We corrected this.

Dear Dr. Barau,

Thank you for the submission of your revised manuscript. We have now received the enclosed reports from the referees, and I am happy to say that both support the publication of your ms now. Referee 1 has one more suggestion that I would like you to incorporate before we can proceed with the official acceptance of your manuscript.

A few editorial requests will also need to be addressed:

- Please add up to 5 keywords to your ms file.
- Please place the Data Availability Section to before the Acknowledgements.
- Please correct the conflict of interest subheading to "Disclosure and Competing Interests Statement".
- Several datasets are uploaded in one zip folder, but each dataset needs to be uploaded as a separate file please.
- Please remove the figure legends for the Appendix figure from the ms file.
- The Reagents & Tools table should be provided as a word file. A downloadable templates (.docx) for the Reagents and Tools Table can be found in our author guidelines: <
<https://www.embopress.org/page/journal/14693178/authorguide#manuscriptpreparation>>.
- Our routine image analysis of to-be-accepted ms detected some potential image aberrations in Appendix Figure S8 around the edges of the image. Can you please check and send us source data for this image.

Figure Legends - Comments

- 1. Please define the annotated p values ****/***/**/* as well as provide the exact p-values for the same in the legends of figures 2C, EV2 C as appropriate/reasonable.
- Please indicate the statistical test used for data analysis in the legends of figures 1C, D; 3A, B; 5A, C; EV5C, E, F.
- Please note that the box plots need to be defined in terms of minima, maxima, bounds of box and whiskers, and percentile in the legends of figures 2C, EV2 C.

I would like to suggest some minor changes to the abstract that needs to be written in present tense. Do you agree with this:

Silencing evolutionary young retrotransposons by cytosine DNA methylation is essential for spermatogenesis, as failure to methylate their promoters leads to reactivation, meiotic failure, and infertility. How retrotransposons reactivate in the absence of DNA methylation is poorly understood. We show that upon defective DNA methylation distinct retrotransposon families display unique expression patterns and chromatin landscapes during mouse spermatogenesis. We find that their reactivation in meiotic spermatocytes correlates with the loss of bivalent H3K4me3-H3K27me3 chromatin marks. Through proteomics and chromatin profiling, we identify NRF1 as a DNA methylation-sensitive transcription factor that transactivates unmethylated retrotransposons. Conditional germline knockout of Nrf1 in the absence of DNA methylation rescues the silencing of the most mutagenic retrotransposon in mice, namely Intracisternal A-particle or IAP. Our findings reveal that chromatin modifications together with a DNA methylation-sensitive transcription factor regulate retrotransposon expression in the absence of DNA methylation in spermatogenesis, revealing a mechanism by which retrotransposons proliferate in the germline after evading DNA methylation-based silencing.

EMBO press papers are accompanied online by A) a short (1-2 sentences) summary of the findings and their significance, B) 2-3 bullet points highlighting key results and C) a synopsis image that is exactly 550 pixels wide and 200-600 pixels high (the height is variable). The synopsis image should provide a sketch of the major findings, like a graphical abstract. Please note that text needs to be readable at the final size. Please send us this information along with the final manuscript.

Referee #1:

The authors have convincingly answered my requests, and I think that the manuscript is now clearer and the signal is easier to interpret. Particularly useful are the NRF vs IgG control plots, which showcases the difference on Dnmt3c KO.

I do think that the small schematic regarding global methylation levels in the germline could be incorporated in Figure 1a (even with "approximate" numbers). I agree with the authors that the meiosis demethylation is subtle and probably shouldn't have much impact here, and there's no need to repeat WGBS done previously.

Referee #3:

The ms is suitable for publication in EMBO reports without revision.

All editorial and formatting issues were resolved by the authors.

Joan Barau
Institute of Molecular Biology - IMB
Ackermannweg 4
Mainz, Rhein-Pfalz 55128
Germany

Dear Joan,

I am very pleased to accept your manuscript for publication in the next available issue of EMBO reports. Thank you for your contribution to our journal.
